

# The impact of melt ponds on summertime microwave brightness temperatures and sea ice concentrations

S. Kern[1], A. Rösel[2], L. T. Pedersen[3], N. Ivanova[4], R. Saldo[5], and R. T. Tonboe[3,a]

[1]Center for Climate System Analysis and Prediction CliSAP, Hamburg, Germany
[2]Norsk Polar Institute, Tromsø, Norway
[3]Danish Meteorological Institute, Copenhagen, Denmark
[4]Nansen Environmental and Remote Sensing Center NERSC, Bergen, Norway
[5]Danish Technical University-Space, Copenhagen, Denmark
[a]currently at: Finish Meteorological Institute, Helsinki, Finland

Received: 1 November 2015 – Accepted: 9 December 2015 – Published: 15 January 2016

Correspondence to: S. Kern (stefan.kern@uni-hamburg.de)

Published by Copernicus Publications on behalf of the European Geosciences Union.

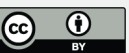

TCD

doi:10.5194/tc-2015-202

Melt ponds impact ice concentration retrieval

S. Kern et al.

## Abstract

The sea ice concentration (SIC) derived from satellite microwave brightness tempera-ture (TB) data are known to be less accurate during summer melt conditions – in the Arctic Ocean primarily because of the impact of melt ponds on sea ice. Using data from June to August 2009, we investigate how TBs and SICs vary as a function of the ice surface fraction (ISF) computed from open water fraction and melt pond fraction both derived from satellite optical reflectance data. SIC is computed from TBs using a set of eight different retrieval algorithms and applying a consistent set of tie points. We find that TB values change during sea ice melt non-linearly and not monotonically as a function of ISF for ISF of 50 to 100 %. For derived parameters such as the polarization ratio at 19 GHz the change is monotonic but substantially smaller than theoretically ex-pected. Changes in ice/snow radiometric properties during melt also contribute to the TB changes observed; these contributions are functions of frequency and polarization and have the potential to partly counter-balance the impact of changing ISF on the observed TBs. All investigated SIC retrieval algorithms overestimate ISF when using winter tie points. The overestimation varies among the algorithms as a function of ISF such that the SIC retrieval algorithms could be categorized into two different classes. These reveal a different degree of ISF overestimation at high ISF and an opposite development of ISF over-estimation as ISF decreases. For one class, correlations be-tween SIC and ISF are ≥ 0.85 and the associated linear regression lines suggest an exploitable relationship between SIC and ISF if reliable summer sea ice tie points can be established. This study shows that melt ponds are interpreted as open water by the SIC algorithms, while the concentration of ice between the melt ponds is in general being overestimated. These two effects may cancel each other out and thus produce seemingly correct SIC for the wrong reasons. This cancelling effect will in general only be 'correct' at one specific value of MPF. Based on our findings we recommend to not correct SIC algorithms for the impact of melt ponds as this seems to violate physical

**TCD**

doi:10.5194/tc-2015-202

**Melt ponds impact ice concentration retrieval**

S. Kern et al.

Discussion Paper | Discussion Paper | Discussion Paper | Discussion Paper

principles. Users should be aware that the SIC algorithms available at the moment retrieve a combined parameter presented by SIC in winter and ISF in summer.

# 1   Introduction

Melt ponds are irregularly shaped puddles of melt water on top of sea ice. They form as a consequence of summer melt of the snow cover on sea ice and the sea ice itself and insufficient draining. The areal fraction, size and depth of melt ponds is determined by the length and characteristics of the melting season, the snow depth at the beginning of melt and sea ice topography and type (Perovich and Polashenski, 2012; Petrich et al., 2012; Perovich et al., 2002). The water inside melt ponds is usually fresh. Even when the melt pond has melted through the ice and is in direct contact with the ocean the melt pond water salinity is substantially lower than that of the ocean. Depending on stage of melt and ice type melt ponds can cover up to between 50 and 90 % of the ice surface (Perovich et al., 2011; Eicken et al., 2004; Yackel and Barber, 2000). The development of melt ponds from initial formation until freeze-up is described in Polashenski et al. (2012) and Perovich et al. (2002).

The presence of melt ponds on the sea ice surface lowers the albedo from about 0.8 without melt ponds to values around 0.5 or even lower with melt ponds (Perovich, 2003, 1996). The increased absorption of incoming shortwave radiation in the presence of melt ponds amplifies heat uptake and accelerates melt (Curry et al., 1995). Thus, melt ponds have an effect on the ice albedo feedback mechanism (e.g. Hudson, 2011), on the sea ice melt rate, the light transmission through the ice (e.g. Light et al., 2008), on Earth's radiation balance (e.g. Maslanik et al., 2007), and even on the aerodynamic surface roughness and thus boundary layer structure and momentum exchange (Lüpkes et al., 2013).

In advanced thermodynamic and dynamic sea ice models integrated in coupled ocean and atmospheric Earth system models or in ocean models used for short term forecast melt ponds are treated separately from open water, cracks, and leads in be-

**TCD**

doi:10.5194/tc-2015-202

**Melt ponds impact ice concentration retrieval**

S. Kern et al.

**TCD**

doi:10.5194/tc-2015-202

**Melt ponds impact ice concentration retrieval**

S. Kern et al.

tween the ice floes with direct contact between the ocean and the atmosphere. In model terminology the area of ice surface covered by melt ponds is called melt pond fraction (MPF), and the fraction of ice covering the ocean (including melt ponds) is called the sea ice concentration (SIC).

This study was initiated in the ESA SICCI project, a project about the construction of sea ice climate records. When constructing climate records it is very important to use a consistent methodology from season to season and year to year and to avoid sensitivity to noise in general and in particular it is important to avoid sensitivity to noise, which is varying with the changing climate. When using satellite microwave radiometer data to derive the sea ice area or extent one such climate sensitive noise parameter are melt ponds. Therefore, we are particularly interested in how melt ponds are represented in microwave radiometer sea ice concentration data which is widely used for sea ice climate records.

Melt ponds are pools of water on the ice floe surface. The penetration depth into liquid water at the microwave frequencies that are used for SIC retrieval (here between 6 and 89 GHz) is in the order of one millimeter (Ulaby et al., 1986). Therefore, the optical depth of a few millimeters water layer in a forming melt pond is enough to diminish the microwave signal of the sea ice beneath; and the melt pond has a brightness temperature (TB) signal of open water. It is therefore expected that when microwave radiometer data are used for SIC retrieval the fraction of open water includes the melt ponds in addition to the open water in cracks and leads. This results in a major shortcoming of today's suite of SIC algorithms (Rösel et al., 2012b). During cold conditions i.e. as long as freezing conditions prevail, current SIC algorithms can retrieve SIC as accurately as 2 to 5 % for the near 100 % ice cover (Ivanova et al., 2015, 2014; Andersen et al., 2007; Meier, 2005). However, during melting conditions the retrieval skill is reduced substantially and SIC is biased low compared to the real ice concentration (Ivanova et al., 2013; Comiso and Kwok, 1996; Cavalieri et al., 1990). A systematic study which quantifies this bias in SIC retrieved from satellite microwave radiometry due to melt ponds for the

Discussion Paper | Discussion Paper | Discussion Paper | Discussion Paper

current suite of algorithms has yet not been done and so that is the purpose of this study.

To increase the reliability of numerical models, they need to be initialized and evaluated with measured sea ice parameters, like e.g. SIC. While the SIC is well defined during winter there is an ambiguity between open leads and melt ponds on the surface of the ice floe during summer. This potential bias in the SIC products due to melt ponds is unknown since the fraction of melt ponds cannot be measured separately using microwave data. Therefore it is difficult to carry out a useful evaluation of models with regard to summer SIC and sea ice area. One might argue that if SIC retrieval during summer simply provides the water area (leads and melt ponds) then modelers could be satisfied because with the open water fraction the albedo can be adjusted correctly. Sea ice albedo has been one of the major challenges in the CMIP5 model runs or in general runs with GCMs (e.g. Karlsson and Svensson, 2013; Björk et al., 2013; Gorodetskaya et al., 2008). However, in order to obtain a correct estimate of the summer sea ice volume and in order to have a correct sea ice distribution at the beginning of the freezing season it is desirable to be able to provide modelers with the actual SIC and not just with the net sea ice surface area, i.e. SIC minus MPF, which is what is measured by microwave radiometry. Note also though, that a number of today's numerical sea ice models already include a melt pond scheme (Holland et al., 2012; Flocco et al., 2010; Pedersen et al., 2009), potentially allowing them to compare with or assimilate ice surface fraction.

An investigation of the sensitivity of present day SIC retrieval algorithms to the presence and amount of melt ponds is required in order to quantify the influence of melt ponds on derived SIC and to find out whether any of the current algorithms is able to reproduce the actual SIC or whether an alternative solution is needed.

The easiest way to circumvent the above-described problem would be: (i) a SIC retrieval method which is not sensitive to melt ponds or (ii) a method that allows to retrieve MPF. In case of (i) SIC would be obtained as if there would be no melt ponds on top and only the leads between the sea ice floes would contribute to the open water

**TCD**

doi:10.5194/tc-2015-202

**Melt ponds impact ice concentration retrieval**

S. Kern et al.

fraction identified by the sensor. In the microwave frequency range typically used for SIC retrieval this is not possible because of the small penetration depth into water as mentioned above. The thermal infrared frequency range (the atmospheric window at 8–14 microns) cannot be used because the surface temperature contrast between the melting ice and melt ponds typically is less than a Kelvin and because also the temperature contrast between the melting ice and open water in leads and openings becomes < 2–3 Kelvin at most at the end of summer (Haggerty et al., 2003; Maslanik et al., 2001; Pegau and Paulson, 1999; Lindsay and Rothrock, 1994) and finally at IR wavelengths the penetration depth is even smaller (micrometers) than in the microwave frequency range. IR observations are additionally hampered by the cloud cover when carried out by satellite sensors (Lindsay and Rothrock, 1994). The visible range also suffers from cloud cover but allows discrimination between leads, melt ponds and bare and/or snow covered ice because of their different optical spectral characteristics (e.g. Morassutti and LeDrew, 1996; Tschudi et al., 2001). Daylight is readily available during summer. A number of studies dealt with the derivation of MPF on Arctic sea ice from visible data (e.g. Markus et al., 2003; Tschudi et al., 2008; Zege et al., 2015; Rösel et al., 2012a; Istomina et al., 2015a, b). By using reflectance values observed by the MODerate resolution Imaging Spectroradiometer (MODIS) Rösel et al. (2012a) computed the MPF with a spectral un-mixing approach in combination with a neural network. This method offers as a by-product the SIC (Rösel et al., 2012a) and is used as reference dataset in the present study.

We will describe the MODIS MPF and SIC data set and present a selection of representative SIC data sets derived from satellite microwave radiometry, using different algorithms. We compare the SIC datasets from optical data with the datasets from microwave radiometer data, i.e. brightness temperatures and the derived SIC, in order to quantify the influence of melt ponds on microwave SIC datasets.

Discussion Paper | Discussion Paper | Discussion Paper | Discussion Paper |

**TCD**

doi:10.5194/tc-2015-202

**Melt ponds impact ice concentration retrieval**

S. Kern et al.

Discussion Paper | Discussion Paper | Discussion Paper | Discussion Paper |

**TCD**

doi:10.5194/tc-2015-202

**Melt ponds impact ice concentration retrieval**

S. Kern et al.

## 2 Data and methods

The MODIS melt pond data used in the present study is based on the Collection V005 "MODIS Surface Reflectance daily L2G Global 500 m and 1 km" – product (MOD09GA) which is available on http://reverb.echo.nasa.gov/reverb/. The L2G data are available
on the sinusoidal tile grid used for MODIS L2 data (http://modis-land.gsfc.nasa.gov/MODLAND_grid.html). To cover the entire Arctic, tiles h09vXX to h26vXX with XX = 00, 01, and 02 are used. The MODIS reflectance data are projected together with land and cloud-mask information onto the NSIDC polar stereographic grid with tangential plane at 70° N with a grid resolution of 0.5 km. Subsequently, all re-projected tiles are
used to compose an Arctic mosaic of the bands 1, 3, and 4, which cover the following wavelength bands: 459 to 479, 620 to 670 nm, and 841 to 876 nm. A spectral un-mixing approach is used to classify the fractions of open water (OWF), melt ponds (MPF), and ice (IF, can be either bare or snow covered ice) based on the typical reflectance values of these surface types in the above-mentioned wavelength bands given by Tschudi et
al. (2008). The methodology is explained in more detail together with validation results in Rösel et al. (2012a) and yields the distribution of OWF, MPF, and IF at 0.5 km grid resolution. This data is subsequently averaged onto a NSIDC polar-stereographic grid with 12.5 km grid resolution. Together with OWF, MPF and IF the MPF standard deviation and the number of clear-sky 0.5 km grid cells contributing to each 12.5 km grid cell
are stored in netCDF format. The number of clear-sky grid cells is taken as a measure of the cloud fraction later. The grid resolution of 12.5 km is chosen in accordance to the 8-day MPF data set derived with the same approach but using 8-day composite MODIS reflectance data for years 2000–2011; this data set is available at the Integrated Climate Data Center (ICDC, http://icdc.zmaw.de). The MPF, as derived here and available
from ICDC is the fraction of melt pond area relative to the SIC. The OWF is used as an independent measure of the SIC by computing 100 % – OWF = MODIS SIC (MSIC); MSIC includes the sea ice covered by melt ponds.

For the inter-comparison with SIC retrieval algorithms and brightness temperatures from satellite microwave radiometry all MODIS data products of June to August 2009 used in the present study, are averaged onto a 100 km resolution grid. This averaging is required because of the large field-of-view of 43 by 75 km of the 6.9 GHz channel of the Advanced Microwave Scanning Radiometer aboard the Earth Observation Satellite (EOS) AQUA (AMSR-E). The region for which these MODIS data products are available is shown in (Fig. 1).

We carried out a bias correction for the daily MPF and MSIC data. This is motivated by an initial comparison of AMSR-E TBs measured at 6.9 GHz with MPF and MSIC and by the results presented in Mäkynen et al. (2014). They hypothesized that the MPF might be biased particularly during early melt by about 5–10 %. The in-situ obser­vations presented in Mäkynen et al. (2014) for 2009 led us to the conclusion that the MPF was still zero north of Greenland during the first two weeks of June 2009 while SIC was 100 %. Melt onset dates given in Perovich et al. (2014) support our conclu­sion. Figure 2 presents histograms of MPF (a) and MSIC (b) for latitudes north of 82° N before the bias correction. Four different histograms are shown in each plot, taking into account all days in June 2009 before 7, 9, 11, and 13 June, respectively. MPF peaks at 8 % without any grid cells with MPF < 4 %. MSIC peaks at 97 % without any grid cell with MSIC = 100 %. Taking the results of Mäkynen et al. (2014) as well as the dis­cussion of uncertainty sources in Rösel et al. (2012a) into account we implemented a bias correction. The MPF was subtracted globally by 8 % and the resulting few negative MPF values were set to zero. 3 % was added to the MSIC globally and cases where MSIC > 100 % after this operation were set to 100 %.

The bias-corrected MPF and MSIC data are subsequently used to compute the net SIC or, as we call it henceforth, the ice surface fraction (ISF) within the respective 100 km grid cell. This is the fraction of the 100 km grid cell with ice which is not cov­ered by melt ponds and hence can be identified by microwave radiometry as ice. In order to mitigate the influence by variations in SIC we only use 100 km grid cells with MSIC > 90 %. In order to minimize cloud cover influence we only use 100 km grid cells

**TCD**

doi:10.5194/tc-2015-202

**Melt ponds impact ice concentration retrieval**

S. Kern et al.

**TCD**

doi:10.5194/tc-2015-202

**Melt ponds impact ice concentration retrieval**

S. Kern et al.

with a cloud cover $< 5\%$; using $0\%$ would reduce the total number of valid 100 km grid cells to less than 100 for the entire data set investigated while with $< 5\%$ we have about 16 500 grid cells. It cannot be excluded that during the period of investigation (June–August 2009) clouds were present in almost every of the 100 km × 100 km grid cells covered by at least 90 % ice. However, it is more likely that the large difference in the number of valid grid cells between applying a cloud-cover threshold of 0 or of 5 % is caused by difficulties of cloud identification over bright surfaces such as sea ice as has been reported elsewhere. We suggest therefore to consider this 5 % as a further measure of the accuracy of both MPF and OWF and hence also ISF. Note that we use the absolute MPF instead of the relative one, i.e., we do not weigh the MPF with the SIC but use the MPF relative to the grid cell area.

MPF and ISF data are co-located with TBs measured by AMSR-E at frequencies of 6.9, 10.7, 18.7, 23.8, 36.5 and 89.0 GHz. The AMSR-E data used are swath data resampled to the 6.9 GHz resolution and taken from the AMSR-E/Aqua L2A Global Swath Spatially-Resampled Brightness Temperatures data set, version 2: http://nsidc.org/data/docs/daac/ae_l2a_tbs.gd.html, (Ashcroft and Wentz, 2013). For our comparison we used TB of the frequencies 6.9, 18.7, 36.5 and 89.0 GHz, named 6, 19, 37 and 89 GHz, henceforth. Data from all AMSR-E passes of the same day as the MODIS data are included, and footprints with a center within 5 km of the center of a MODIS 100 km cell. AMSR-E sampling is approximately every 10 km so this gives us approximately 1 data point from each AMSR-E pass.

In order to investigate the sensitivity of the various SIC algorithms considered in the SICCI (European Space Agency Climate Change Initiative – Sea Ice) project we computed SIC from these co-located TBs. The full suite of SIC algorithms used is documented in the SICCI project reports: PVASR (Ivanova et al., 2013) and ATBD (Ivanova et al., 2014). In the present study we focus on a selected number of different (representative) types of SIC algorithms and we do not include all SIC algorithms where some duplicate methodology. The selected algorithms are summarized in Table 1.

Most SIC algorithms can be categorized into four types: (1) algorithms which use TBs at one polarization and one frequency (e.g. One_channel 6H) (S = Single); (2) algorithms which use TBs at different frequencies but the same polarization such as the open water and intermediate SIC part of the Comiso Bootstrap algorithm (Bootstrap_f) (F = Frequency); (3) algorithms which use TBs at different polarizations but the same frequency such as the ice part of the Comiso Bootstrap algorithm (Bootstrap_p), ARTIST Sea Ice algorithm (ASI) and Near 90 GHz linear approach (N90 or Near90_lin); P = Polarization), (4) algorithms which use information from at least two frequencies and/or polarizations like the NASA-Team algorithm (NASA_Team), the enhanced NASA-Team algorithm (NT2), and the Bristol algorithm (Bristol) (M = Mixed).

The next advanced type of algorithm would be a hybrid algorithm combining two or more of the above-mentioned types like the Eumetsat OSI-SAF algorithm combining Bristol and Bootstrap_f (Eastwood et al., 2015). Results from that algorithm are not shown here however, because these are almost identical to the algorithm they use at high SIC (Bristol). We also note that in the Arctic the Bootstrap algorithm is used as a hybrid of both realizations. Bootstrap_p is used for high ice concentrations, i.e. SIC > 90 %. Bootstrap_f is used for all remaining SICs (Comiso, 1986; Comiso et al., 1997, 2009). As we – similar to what has been done by Comiso (1986) and Comiso et al. (1997) – aim to develop an optimal SIC retrieval algorithm within the SICCI project we analyzed both realizations separately. This will be taken into account in the discussion of the results.

Our data set covers areas with multiyear ice (MYI) and first-year ice (FYI). In order to investigate whether the TBs and retrieved SIC change differently for these two ice types we included information about the distribution of MYI and FYI into our analysis. We used the Arctic sea ice age data set (Maslanik et al., 2011; Tschudi et al., 2015; http://icdc.zmaw.de/seaiceage_arctic.html?&L=1) to define which part of our area of interest is covered by MYI. We averaged the ice age data set available at 12.5 km grid resolution onto the same grid with 100 km grid resolution used for the MODIS data (see above). In most of the following figures we show data from all grid cells regardless of ice

**TCD**

doi:10.5194/tc-2015-202

**Melt ponds impact ice concentration retrieval**

S. Kern et al.

type. For Figs. 7 and 8, however, only grid cells with an average age of ≥ 4.0 years are considered MYI. This procedure is applied for each day and results in the distribution shown in Fig. 1. In the ice age retrieval method (Maslanik et al., 2011; Fowler et al., 2003) the age of a grid cell is incremented by one year at the end of summer even if the coverage of the remaining oldest ice is only 15 %. Consequently, at the end of summer a grid cell with 85 % FYI and 15 % 4-year old MYI will be set to a grid cell with 5 years old MYI. The ice age product is hence likely to be biased towards too old ice. Secondly, a considerable number of the 12.5 km grid cells in the 7 × 7 grid cell boxes used to compute the ice age at 100 km grid resolution could contain FYI. In order to minimize the influence of FYI and ice which is actually younger than indicated we chose the threshold of 4 years. Grid cells with an average sea ice age between 1.0 and 4.0 are thus not considered as MYI in Fig. 8 or in general when we write about MYI in the present paper.

## 2.1 Uncertainties of the MODIS data sets

MPF and ISF = 100 − (OWF + MPF) as used in this work may suffer from more uncertainties and biases than those we have already corrected (see above). One of the known problems is the different spectral responses of melt ponds on MYI or FYI caused by the different ice structure and thickness underneath the pond and by the tendency for deeper ponds on MYI than on FYI (e.g. Morassutti and LeDrew, 1996). Potentially this could be taken into account in the retrieval algorithm. However, only 3 different channels are used in the MPF retrieval algorithm (Rösel et al., 2012a) and it is therefore difficult to discriminate between more than three surface types.

Another issue is the cloud masking. Even though a state-of-the-art cloud masking scheme has been applied to the MODIS reflectance data before being used in the MPF retrieval algorithm (Rösel et al., 2012a), there is still a substantial number of misclassified grid cells in the retrieved daily OWF, MPF and MSIC data. It has been demonstrated that even with a multi-channel instrument such as MODIS cloud classification is a challenge over sea ice (Chan and Comiso, 2013; Karlsson and Dybbroe,

**TCD**

doi:10.5194/tc-2015-202

**Melt ponds impact ice concentration retrieval**

S. Kern et al.

**TCD**

doi:10.5194/tc-2015-202

**Melt ponds impact ice concentration retrieval**

S. Kern et al.

2010). In the present study we only use data where at least 95 % of each 100 km grid cell is cloud free to minimize this uncertainty. This reduces the number of misclassified grid cells efficiently but does not eliminate them completely.

In addition, misclassification of one surface type, e.g. MPF, impacts the classification of the other two surface types as well because the sum of the fraction of all three surface types has to equal 1 (Rösel et al., 2012a).

Therefore, there are considerably uncertainties in the MODIS melt pond dataset and we are not attempting to quantitatively "tune" the SIC retrieval algorithms towards more realistic SIC values by means of the difference between ISF and AMSR-E SIC. We rather suggest taking our observations as a guideline along which both optimization and interpretation of MPF and MSIC retrieval and optimization of passive microwave SIC retrieval during summer, e.g. from AMSR-E, could be carried out.

## 3 Results

### 3.1 Development of MPF and SIC as seen by MODIS

Figure 3 shows the development of MSIC, ISF, total open water fraction (OWF + MPF), and MPF during June–August 2009. Note that these are all bias-corrected values (see Fig. 2 and discussion to that). For each day and each co-located data point all four parameters are shown in the scatterplot. The data are shown as is and no further averaging is carried out. There is no distinction between MYI and FYI in this figure. Gaps in the time series and the varying number of data points shown result from varying cloud cover and the decreasing sea ice cover from June to August. Only grid cells with MSIC > 90 % are shown; the number of grid cells fulfilling this criterion is decreasing with progressing melt season. We neglect periods with substantial scatter in the MPF related parameters. These are days 10–18, 60–65 and 84–88.

MPF in Fig. 3 remains near zero during the first 2–3 weeks. Then the MPF starts to increase, first slowly: days 20–30, then rapidly: days 30–45. After a short plateau where

**TCD**

doi:10.5194/tc-2015-202

**Melt ponds impact ice concentration retrieval**

S. Kern et al.

MPF remains near 35 % it first declines rapidly to about 20 % at days 55–60 and then more slowly to about 15 % until the end of our study period (31 August). Throughout June MSIC is close to 100 % (until day 30), and then there is more variability around 90–95 % towards end of July (after day 55). The other two parameters, total open water fraction and ISF are linked to MSIC and MPF, and add up to 100 %.

In the following two sections we focus on MODIS ISF. We use ISF instead of MSIC because melt ponds in the passive microwave data is expected to show up as areas of open water. The penetration depth of microwave radiation into water is of the order of a few millimeters at the frequencies used by the algorithms listed in Table 1 (Ulaby et al., 1986). Hence, in theory, melt ponds on sea ice influence TBs in the same way as open water in leads and openings between the ice floes does. The difference in water salinity can be neglected at the frequencies which are used here (Ulaby et al., 1986). We will show first how TBs and derived ratios used by the algorithms listed in Table 1 vary as function of MODIS ISF (Figs. 4 to 6). Then we will show how the SIC obtained with the algorithms listed in Table 1 compares to MODIS ISF (Figs. 7 and 8). Because MPF and hence ISF varies most during July (see Fig. 3) we focus on data from the month of July.

## 3.2 AMSR-E observations in comparison to MODIS ice surface fraction

Figure 4 shows scatterplots of AMSR-E TB at the frequencies 6, 19, 37, and 89 GHz vs. MODIS ISF (henceforth named ISF) for June to August (top row) and July (bottom row). At 6 and 19 GHz, TBs increase with increasing ISF (Fig. 4a, b, e, f). At 37 GHz it is difficult to discriminate between TB increase or decrease with increasing ISF (Fig. 4c, g). At 89 GHz mean TB tends to decrease with increasing ISF (Fig. 4d, h) because sea ice TB < open water TB at this frequency. The general development of TB with increasing ISF is similar for the entire period (Fig. 4a–d) and for July only data (Fig. 4e–h). We note that other effects than melt ponds may influence the TB. Such effects are, e.g., melt-freeze cycles including the varying amount of liquid water in the snow, the transition from snow covered ice to bare ice or vice versa, changes in the emitting layer

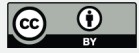

temperature, and metamorphosis of the snow layer and the surface ice layers including snow grain size growth and porosity of the ice. Therefore, we should not necessarily expect a simple relationship between SIC and ISF.

At 6 GHz and also at 19 GHz, TBs scatter at one ISF value by 15 to 20 K (Fig. 4a, b, e, f). This is much less than at 37 and 89 GHz where TBs at one ISF value can range over 50 K (Fig. 4c, d, g, h). In general, the scatter in TBs at one ISF value decreases when considering only July data. During July melting conditions usually have commenced everywhere while during June surface conditions are more variable with the co-existence of still cold, moderately warm pre-melt, and mature melt conditions.

At ISF ≈ 100 % TBs are about 20 K below TBs at ISF ≈ 95 % for both polarizations at 19 GHz (Fig. 4b, f). This is not the case at 6 GHz (Fig. 4a, b). At 37 and 89 GHz this TB decrease between ISF ≈ 95 % and ISF ≈ 100 % is much larger: 50–60 K and TBs take values of 200 and 185 K at V- and H-polarization, respectively (Fig. 4g, h). We explain this with the colder temperatures for some of the data at ISF ≈ 100 %. We should therefore expect a peak of TBs at 80–95 % ISF at which temperatures are most likely at the melting point and thus MYI signatures have changed to the near blackbody signature of wet snow.

Basically all results shown in Fig. 4 are also valid when limiting the data set to MYI (not shown). The change from an increase in TB with increasing ISF at frequencies 6 to 37 GHz to a decrease in TB with increasing ISF at 89 GHz is more pronounced for MYI, as expected.

Figure 5 illustrates the reversal of the change in TB with increasing ISF between low and high frequencies more clearly. At 6 GHz (Fig. 5a, e) TBs peak at low values for low ISF and at high values for ISF > 95 %. In contrast, at 89 GHz (Fig. 5d, h) TBs peak at high values for low ISF and at low values for ISF > 95 %. There is a considerably larger change of TB at H- than at V-polarization for 6 GHz (Fig. 5a, e) while almost no difference between polarizations is evident at 89 GHz (Fig. 5d, h). This is expected since at H-polarization TBs over water are much smaller than at V-polarization – particularly for the lower frequencies shown.

**TCD**

doi:10.5194/tc-2015-202

**Melt ponds impact ice concentration retrieval**

S. Kern et al.

Some of the algorithms listed in Table 1 use ratios of TBs at different polarizations and/or frequencies, e.g. the polarization ratio (PR): the difference of vertically minus horizontally polarized TBs divided by their sum, or the gradient ratio (GR): the difference between TBs at two frequencies but same polarization divided by their sum. Relevant in the present study are PR at 19 GHz (PR19) and GR between 37 and 19 GHz, V polarization (GR3719); these are used in the NASA Team and NT2 algorithms. We include PR at 37 and 89 GHz (PR37, PR89) as well. Figure 6 illustrates: PR19 apparently starts to increase once ISF falls below ≈ 75 % (Fig. 6b); PR37 seems to be less sensitive to ISF and starts to increase once ISF falls below 65 % (Fig. 6c); PR89 is rather independent of ISF (Fig. 6d). GR3719 increases considerably with decreasing ISF already in the range of 75 % < ISF < 95 %. For smaller ISF GR3719 stays relatively constant at around zero (Fig. 6a).

Any quantitative conclusion from Figs. 5 and 6 would require a more accurate ISF. Because both MPF and MSIC, the two input variables for ISF, are subject to a number of uncertainty sources as has been discussed at the end of the "Data and methods" section, we recommend to not over-interpret the findings of these two figures. Note also that bin ISF < 55 % contains substantially less data than all other bins. However, Figs. 5 and 6 illustrate once again that there are substantial differences in the sensitivity of TBs at different frequencies as measured by AMSR-E to co-located MODIS ISF data (Fig. 4) and that these differences are not necessarily mitigated by using derived parameters such as the PR or the GR (Fig. 6). A number of other factors than ISF are influencing TBs in this investigation and may explain some of the observations. At high ISF the ice may (still) be colder and exhibit MYI signatures whereas at lower ISF the ice is more likely to be exhibiting the near blackbody signature of wet snow, until the ice becomes bare, which again may cause a change in the TB. The different wavelengths and hence penetration depths into snow and ice contribute also to the differences observed. In addition it should be remembered that TB of water is quite different at the different frequencies and polarizations and is also subject to the influence of emissivity changes due to surface roughening by the surface wind.

**TCD**

doi:10.5194/tc-2015-202

**Melt ponds impact ice concentration retrieval**

S. Kern et al.

Discussion Paper | Discussion Paper | Discussion Paper | Discussion Paper |

Discussion Paper | Discussion Paper | Discussion Paper | Discussion Paper |

**TCD**

doi:10.5194/tc-2015-202

**Melt ponds impact ice concentration retrieval**

S. Kern et al.

### 3.3 AMSR-E sea ice concentration compared to MODIS ISF

Figure 7 shows scatterplots of SIC and ISF for June to August 2009 (grey symbols) and July only (black symbols). Black symbols are overlying grey symbols. All SIC computed with the different algorithms are derived from the same AMSR-E TB dataset. The algorithms which are listed in Table 1 are using a consistent set of winter tie points (Ivanova et al., 2015, 2014). Figure 8 contains a subset of Fig. 7 and shows data pairs for MYI with an average ice age of 4 years or older.

All AMSR-E SIC algorithms in Fig. 7 overestimate the ISF. For all algorithms, the majority of the SIC – ISF data pairs is situated above a 1-to-1 regression line (not plotted, would connect $(0,0)$ with $(100,100)$). This applies in particular when considering data from July only (black symbols) with maximum advance of melt (see Fig. 3). When taking into account June and August as well, then all algorithms except Bootstrap_f and Bristol (Fig. 7a and b) show a considerable fraction of data pairs below the 1-to-1 line – particularly at ISF > 80 %, e.g. Bootstrap_p and Near90_lin (Fig. 7e and f).

The largest over-estimation of the ISF is from Bootstrap_f (Fig. 7a) where SIC gets as high as 140 % for ISF > 85 %. For Bristol (Fig. 7b) the SIC peaks between 120 and 130 %. SIC from One_channel 6H, NASA_Team, and Near90_lin (Fig. 7h, c and f) peak between 110 and 120 %. The lowest over-estimation of ISF is from Bootstrap_p (Fig. 7e).

ASI and NT2 cannot be considered here because both algorithms cut off SIC at 102 and 100 %, respectively (Fig. 7g and d). These two algorithms also show the smallest correlation between SIC and ISF: 0.38 for NT2 and 0.42 for ASI and are therefore not considered in the discussion of Figs. 7 and 8. Ivanova et al. (2015) showed that this behavior is due to over-estimation of SIC at higher SIC values by these algorithms, which combined with the cut-off at 100 % results in this lack of (expected) correlation. It is noted that the high estimates of SIC can to some extent be adjusted with the tie points except for the NT2 and ASI algorithms where the tie points are an implicit part of the algorithm.

Common to the other six SIC data sets is a regression line through the July only SIC vs. ISF data points (black solid lines in Fig. 7) with a slope steeper than 1. The slope varies between 1.13 for Bootstrap_p (Fig. 7e) and 1.42 for Bootstrap_f (Fig. 7a), see also Table 2. These slopes provide the envelope for the slopes obtained for SIC of the other algorithms. Two algorithms have regression line slopes close to 1.2 (NASA_Team and Near90_lin) and two algorithms have a slope close to 1.3 (Bristol and One_channel 6H). This suggests a systematic shift towards higher SIC values, which is linearly related to ISF, for all SIC retrieval algorithms investigated.

Figure 8 reveals similar relationships between SIC and ISF using only grid cells with MYI with an average age of 4 years or more. The statistical parameters derived, e.g. slope, correlation and RMSD, are similar to those shown in Fig. 7 (see Table 2).

Highest correlations are obtained by Bootstrap_f: 0.85 and 0.91 for all and only the MYI grid cells, respectively, followed by Bristol: 0.85 and 0.86 and One_channel 6H: 0.81 and 0.86. For all other algorithms – except NASA_Team – correlations are $\leq 0.5$. With regard to RMSD again Bootstrap_f and Bootstrap_p provide the envelope within which RMSD values of the other algorithms lie: the largest: 33.3 % (33.4 %), and the smallest: 15.3 % (16.7 %) RMSD values, respectively, for all (only MYI) grid cells. The other algorithms have RMSD values around 25 % (Bristol and One_channel 6H) or around 20 % (NASA_Team and Near90_lin) (Table 2).

One can group the eight algorithms into three classes: class I: algorithms with a relatively steep slope ($\geq 1.3$) but a regression line which would also naturally intersect close to $(0, 0)$, a high correlation ($> 0.8$) and relatively little scatter around the regression line: Bootstrap_f, Bristol, One_channel 6H; class II: algorithms with a regression line slope closer to 1 but a regression line which would not intersect at $(0, 0)$, considerably lower correlation than algorithms from class I, and more scatter of data pairs around the regression line: Bootstrap_p, Near90_lin, NASA_Team; class III: algorithms without any clear relationship between SIC and ISF because SIC is cut off at or close to 100 %: ASI, NT2. The algorithms of class III were found to substantially over-estimate SIC in the range 75 to 100 % (Ivanova et al., 2015).

**TCD**

doi:10.5194/tc-2015-202

**Melt ponds impact ice concentration retrieval**

S. Kern et al.

All algorithms used here (Table 1) were compared to independent reference SIC data (Ivanova et al., 2015). All but ASI and NT2 revealed a bias < 5 % for SIC ≥ 75 % during winter. High SIC areas during summer could not be considered by Ivanova et al. (2015) due to the lack of high-quality independent SIC observations.

## 4 Discussion

### 4.1 About TB, ISF, SIC and tie points

As detailed earlier we expect SIC derived from microwave radiometry to be similar to the ISF independently of OWF, MPF and frequencies used. At the microwave frequencies typically used for SIC retrieval (see Table 1) penetration depth into water is of the order of a few millimeters (Ulaby et al., 1986). Therefore we cannot discriminate between open water in leads/openings and melt ponds. Both surface types should be interpreted as open water.

For the present paper all algorithms have been applied to TBs measured during summer conditions (June to August 2009) using the same set of tie points, derived from winter observations similarly to Ivanova et al. (2015). Tie points are typical TB values from areas of known surface composition: open water or ice. For SIC retrieval one usually employs an open water tie point for SIC = 0 % ($TB_{OW}$) and a sea ice tie point for SIC = 100 % ($TB_{ICE}$). For sea ice in the Arctic one can, in addition, discriminate between FYI ($TB_{FYI}$) and MYI ($TB_{MYI}$). The set of tie points used by Ivanova et al. (2015) and in the present paper is valid for winter conditions. During summer, snow and ice surface properties change due to melting and presence of melt ponds. Melt onset varies in time and space from year to year. Melt progress varies regionally as well and the first appearance of melt ponds and/or bare ice is a complex function of meteorological conditions, snow depth and topography and ice type and topography.

In Fig. 4, TBs at ISF = 100 % should be close to the red and green symbols denoting $TB_{FYI}$ and $TB_{MYI}$, but they are not. Observed TBs at all frequencies are shifted towards

**TCD**

doi:10.5194/tc-2015-202

**Melt ponds impact ice concentration retrieval**

S. Kern et al.

higher values than $TB_{FYI}$ and $TB_{MYI}$, when the full data set (June to August) is used. This shift could be explained e.g. by increased wetness of the ice/snow surface, which causes TBs to increase and an over-estimation of SIC per areal unit, and therefore larger areas of ice lead to larger overestimation. The TB increase with snow wetness

during melt onset has been demonstrated, e.g., for TBs at 37 GHz by Garrity (1992).

When limiting the data set to July the TB shift towards higher values only applies to 6 GHz. TBs at 19 GHz for ISF ≈ 100 % agree with $TB_{ICE}$ at least for FYI, TBs at 37 GHz fall between $TB_{FYI}$ and $TB_{MYI}$, while TBs at 89 GHz agree best with $TB_{MYI}$. This change in the different shift between summer and winter TBs during summer can

be explained with the fact that the increase in wetness of the ice/snow surface is only one of the physical processes occurring during summer melt. Melt-freeze cycles, melt pond development, snow grain size increase, and the various changes occurring at the surface of the bare ice later in the melt season influence TBs in a more complex way than during the beginning of the melt season.

For SIC estimation we hypothesize a linear relationship between TB and SIC. Accordingly, SIC should increase along a line connecting $TB_{OW}$ and $TB_{ICE}$ at the same polarization. At 6 GHz, for example, this line would have a positive slope for both FYI and MYI (see also Table 3). To obtain SIC with a simple linear approach, observed TBs would need to fall between $TB_{OW}$ and $TB_{ICE}$ at the same polarization. This is clearly not

the case as is demonstrated for 6 GHz and the One_channel 6H algorithm in Figs. 5, 7h and 8h). At the other three frequencies the relationship between TBs, ISF, and $TB_{FYI}$ and $TB_{MYI}$ is more complex and non-linear, particularly at 37 and 89 GHz (Fig. 5) for which at least during July 2009 TBs measured over sea ice take values in basically the entire TB-space given by the winter tie points – as a function of ISF.

These findings suggest that SIC retrieval will be biased if using winter tie points. It follows furthermore that the bias depends on ice type. Algorithms employing TBs at 37 GHz, either alone or in combination with TBs at 19 GHz, e.g. Bootstrap_f, Bootstrap_p, and Bristol, over-estimate ISF considerably (Figs. 7, 8). For ISF = 70 % SIC takes values up to 110, 95, and 100 %, respectively, for these algorithms, and for ISF

**TCD**

doi:10.5194/tc-2015-202

**Melt ponds impact ice concentration retrieval**

S. Kern et al.

= 90 % SIC takes values up to 140, 110, and 130 %, respectively. Plots (d) and (g) of Figs. 7 and 8 illustrate that generally limiting the SIC to 100 % does not solve the problem because in that case the natural variability of the actual ice cover is not represented by the retrieved SIC.

Some of the algorithms shown in Table 1 employ TB ratios, i.e. PR19 like NASA_Team and NT2. SIC values start to exceed 100 % at ISF $\geq$ 70 % for NASA_Team (Fig. 7c) and at ISF $\geq$ 60 % for NT2 (Fig. 7d), also for MYI with an average age of 4 years or older (Fig. 8c, d). For ISF > 85 %, PR19 histograms peak at the FYI tie point and then move towards larger PR19 values with decreasing ISF – and hence towards $TB_{OW}$ (Fig. 6b). However, this shift towards larger PR19 values is less pronounced than theoretically expected. The PR19 open water tie point is 0.26 and FYI and MYI tie points are 0.030 and 0.043, respectively (Table 3) – as computed from the winter TB values in Ivanova et al. (2015). If we assume a linear relationship between 100 % and 0 % FYI we would expect PR19 = 0.122 for SIC = 60 % (ISF = 60 %). However, for bin 55 % < ISF < 65 % PR19 values peak at 0.055 (Fig. 6b, red curve). This is half the value expected and it suggests a considerable over-estimation of SIC for decreasing ISF. For MYI PR19 at SIC = 60 % would be even higher.

Our ISF values might be biased by 10 %. If we use the adjacent ISF bins and hence use SIC = 50 % or SIC = 70 %, then PR19 would range between 0.099 and 0.145. The PR19 values, at which the respective ISF bins peak, would still lie outside this range underlining the potential over-estimation of SIC with decreasing ISF when based on PR19.

In an attempt to check whether PR values during melting conditions differ from the winter ones shown in Fig. 6 we used the MODIS ISF of the period 20 June to 5 July to derive summer tie points from the AMSR-E TBs used in this study. We only considered data with ISF > 97.8 % and TBs at 37 GHz, V-polarization, above 250 K and did not discriminate between different ice types. PR values computed from the selected TBs are given in Table 3 for both winter (Fig. 6) and summer; the summer TBs are also given in Table 3 as well as in Fig. 4, bottom row, purple symbols.

**TCD**

doi:10.5194/tc-2015-202

**Melt ponds impact ice concentration retrieval**

S. Kern et al.

**TCD**

doi:10.5194/tc-2015-202

**Melt ponds impact ice concentration retrieval**

S. Kern et al.

Comparison of the PR values reveals: the summer PR19 lies between winter PR19 tie points for FYI and MYI (Table 3). Hence, our hypothesis that a PR19 difference between summer and winter could explain our findings shown in Fig. 6 is not valid. More investigations with larger, more representative data sets are required in order to figure out whether PR values in summer stay as close to the winter values as suggested by our analysis. If this is going to be confirmed other effects need to be considered in order to explain the apparent difference between theoretically expected and actually observed PR19 during July 2009 in the Arctic Ocean shown in Fig. 6b). Note for completeness that also our summer PR89 is identical to the winter PR89 FYI tie point and hence only 0.003 larger than the winter MYI PR89 tie point, and that only our summer PR37 takes a slightly larger value, exceeding the winter FYI and MYI PR37 tie point values by 0.008 and 0.002, respectively. As has been pointed out in the previous section already, however, the change in ISF is perhaps not the main influencing factor for a change in PR values because of the non-linear response between ice and snow property changes during melt onset and progress on the one hand and the frequency dependent radiometric response to these changes on the other hand.

A straightforward conclusion from the results of the previous paragraphs would be: Sea ice tie points which represent winter-time radiometric sea ice properties are not suited to derive SIC under summer conditions. This is not new. For instance, Comiso et al. (1997) use seasonally varying sea ice tie points in the Comiso Bootstrap algorithm (see also later in Figs. 11 and 12). Eastwood et al. (2015) use a sliding 30-day time window for the sea ice tie point estimation to account for the seasonal variation of sea ice conditions in the Eumetsat OSI-SAF algorithm. Finally, the Eumetsat/ESA SICCI algorithm further enhances the sea ice tie point estimation utilized by the Eumetsat OSI-SAF algorithm by, e.g., using a shorter time window as detailed in the SICCI project report ATBD v2 (Ivanova et al., 2014).

Earlier work has demonstrated that SIC uncertainty (and bias) is larger during summer than during winter (Cavalieri et al., 1990; Steffen and Schweiger, 1991; Comiso and Kwok, 1996). This has been confirmed later by Rösel et al. (2012a, b). In what fol-

**TCD**

doi:10.5194/tc-2015-202

**Melt ponds impact ice concentration retrieval**

S. Kern et al.

lows we will illustrate and discuss by means of the relevant TB-parameter space how TB varies as function of ISF for Near90_lin, NASA_Team, Bootstrap_f and Bootstrap_p algorithms.

The variation in ISF is mainly driven by the MPF since we restrict our analysis to data from July and to grid cells with MSIC > 90 %. Figure 3 reveals that MSIC ≥ 95 % for most of July and most grid cells. Consequently, a grid cell with, e.g., ISF ≈ 70 % contains in most cases less than 5 % open water between ice floes but has a MPF ≈ 25–30 %.

## 4.2  Near90_lin

Figure 9 illustrates how, at 89 GHz, TB values for a typical winter day (light grey and black symbols) compare to TBs during July 2009 (colored symbols). The red dashed line connecting $TB_{FYI}$ and $TB_{MYI}$ (ice line) is almost identical to the blue lines denoting the TB polarization difference (V minus H polarization) computed from $TB_{FYI}$ and $TB_{MYI}$ and parallels the blue line through $TB_{OW}$ (water line). Only a change in TB as function of ISF perpendicular to the ice and water lines affect SIC retrieved with an algorithm using the TB difference at 89 GHz, e.g. ASI and Near90_lin (plots f and g in Figs. 7 and 8). A change in TB as function of ISF parallel to the ice and water lines does not affect SIC. Because $TB_{FYI}$ and $TB_{MYI}$ are located on the ice line any over- or under-estimation in SIC due to changes in ISF are independent of ice type.

TB values associated with ISF ≈ 100 % are mostly located above the ice line. Those associated with ISF < 80 % are located below the ice line. There seems to be a decreasing gradient in ISF perpendicular to the ice line into direction of the water line. TB values associated with ISF ≤ 70 % (MPF ≥ 25 %) are primarily located close to $TB_{FYI}$ and even higher TBs. This applies in particular to the subset of ice with a mean age of four years or older (Fig. 9b) where such TBs are even found on the ice line. Note that Fig. 9b (and corresponding plots in Figs. 10 to 12) is based on about just a quarter (2323) of the points shown in Fig. 9a (7927). Only 120 data points are pure FYI, i.e. had a mean sea ice age of < 1 year. There are also quite some points with ISF ≈ 70 %

**TCD**

doi:10.5194/tc-2015-202

**Melt ponds impact ice concentration retrieval**

S. Kern et al.

located above the ice line (Fig. 9a). Figure 9 illustrates that similar TB polarization differences co-exist for quite different ISF values (see Fig. 6 d). Along the ice line we see ISF $\approx$ 100 % close to TB$_{\text{MYI}}$ (MPF $\approx$ 0 %) and ISF $\approx$ 60 % way up along the ice line towards higher TBs (MPF $\approx$ 35 %).

Note that ISF most likely plays a dominant but not an exclusive role for the observed TB distribution. We need to keep in mind that Fig. 9 – as well as Figs. 10–12 – shows data from an entire month (July). Environmental factors influencing the onset and progress of snow and ice melt like, e.g., surface temperature and down-welling short- and longwave radiation vary over space and time. We need to keep in mind also that we look at the entire Arctic Ocean and hence simultaneously at areas with different snow depth and vertical structure, and different snow and ice surface topography; areas of bare ice co-exist with areas still covered with wet snow and areas where snow might have undergone some metamorphism but are still not yet melting at the beginning of July (see e.g. Perovich et al., 2014). At the end of July, however, coexistence of bare ice and melt ponds is likely to dominate. Hence Fig. 9 – as well as Figs. 10–12 – collects TBs from a suite of variable surface states. It is hence not surprising that we see evidence for an apparently non-linear behavior in the TBs which illustrates the difficulty in potentially deriving seasonally varying tie points for ASI or Near90_lin that is valid for all regions; currently the ASI algorithm is run with static tie points (Kaleschke et al., 2001; Spreen et al., 2008).

## 4.3 NASA_Team

Figure 10 illustrates the NASA_Team tie point triangle for typical winter conditions (black and grey symbols) and illustrates how PR19 and GR3719 derived from AMSR-E TBs measured in July 2009 (colored symbols) change as function of ISF. A considerable number of PR-GR-pairs is located left of the ice line; these are associated with ISF between 80 and 100 %. A cloud of PR-GR-pairs associated with 60 % $\leq$ ISF $\leq$ 80 % is located to the right of the ice line and covers approximately that area in PR-GR space which represents SIC between 80 and 100 %. Near the FYI line, PR-GR-pairs asso-

**TCD**

doi:10.5194/tc-2015-202

**Melt ponds impact ice concentration retrieval**

S. Kern et al.

ciated with ISF ≈ 60 % extend over SIC of 80 to 95 % (Fig. 10b). The ISF associated with the bulk of the PR-GR-pairs shown in Fig. 10a) follows a decreasing gradient almost perpendicular to the ice line. This indicates that the NASA_ Team algorithm sees melt ponds as open water. However, ISF in summer (and using winter tie-points) is over-estimated by between 10 % (at higher ISF) and 30 % (at lower ISF).

For ice with a mean age of 4 years or older (Fig. 10b) PR-GR-pairs spread almost parallel to the ice line towards the FYI tie point and the FYI line. Different ISF values co-exist for one PR19 value. The shift in GR3719 is confirmed by Fig. 6a): While for ISF > 95 % the GR3719 histogram (grey curve) peaks even below the MYI tie point (green dotted line), histograms move towards the FYI tie point (red dotted line) or even farther (red curve for 55 % ≤ ISF ≤ 65 %). The above-mentioned co-existence of different ISF for a particular PR19 value and the smaller than theoretical increase of PR19 with decreasing ISF (see discussion to Fig. 6 b) is, to our opinion, partly responsible for the ISF over-estimation by the NASA_Team algorithm. This over-estimation increases with decreasing ISF. This is also indicated in image (c) of Figs. 7 and 8 where a regression line, which if it were not forced through (0,0), would have a slope < 1. We refer to the discussion of Fig. 9, though, where we mentioned that Figs. 9–12 are integrating over a suite of temporally and spatially changing surface states and hence variations in the properties which determine the radiometric signal of the summer-time sea ice. A changing ISF due to the developing melt pond coverage is only one of the changing surface parameters.

## 4.4 Comiso-Bootstrap frequency mode: Bootstrap_f

Figure 11 illustrates by means of the Bootstrap_f winter tie points and ice line (compare Comiso and Kwok, 1996, Figs. 4 13) superposed onto TB37V – TB19V space for typical winter conditions (black and grey symbols) how AMSR-E TBs measured during July 2009 at 37 and 19 GHz, V polarization (colored symbols) vary as function of ISF. Summer TB values associated with ISF ≈ 100 % are located way above the ice line forming a cloud parallel to it starting at the MYI line at TB19V = 242 K, TB37V = 192 K;

these points are associated with SIC ≈ 140 % (compare Fig. 7a). Further to the right, for TB37V > 210 K TBs start to form an extended cloud with a clearly visible decreasing gradient in the associated ISFs. This gradient follows the direction of a decrease in SIC along the dashed white lines. This clear relationship between Bootstrap_f SIC and ISF

is also indicated by Figs. 7a and 8a and the regression parameters (Table 2).

To the right of the dashed white line in the center, the bands of TBs associated with decreasing ISF are not parallel to the ice line anymore but are getting closer to it. As a result ISF over-estimation becomes smaller the closer we get to the FYI line. In addition, the distance between $TB_{OW}$ and the ice line is smaller along the MYI line than

along the FYI line. This means: the same distance between a TB data pair and the ice line parallel to the dashed white lines translates into a larger SIC change near the MYI line than near the FYI line. Consequently, ISF over-estimation by Bootstrap_f is also a function of the ice type. For ice of an average age of 4 years or older (Fig. 11b) Bootstrap_f over-estimates ISF by 40 % for ISF ≈ 100 %, i.e. for practically melt-pond free

sea ice; the radiometric signature is that of MYI. The more the radiometric signature of the MYI has changed towards that of FYI and/or the more an increase in MPF has caused a decrease in ISF, the smaller becomes the ISF over-estimation by Bootstrap_f; for ISF ≈ 60 % it is just 20 % when using fixed winter tie-points.

## 4.5 Comiso-Bootstrap polarisation mode: Bootstrap_p

Figure 12 illustrates by means of the Bootstrap_p winter tie points and ice line superposed onto TB37V – TB37H space for typical winter conditions (black and grey symbols) how AMSR-E TBs measured during July 2009 at 37 GHz, V and H polarization (colored symbols) vary as function of ISF. Relatively few TBs are located above the ice line and hence at SIC > 100 %. These are mostly associated with ISF > 90 % and

confirm image (e) of Figs. 7 and 8 that for large ISF Bootstrap_p SIC peak at 110 %.

The majority of the summer TBs is situated below the ice line and is mostly associated with 60 % ≤ ISF ≤ 85 %. Similar to Bootstrap_f (Fig. 11a), the distance between $TB_{OW}$ and the ice line is smaller along the MYI line than along the FYI line. TBs asso-

Discussion Paper | Discussion Paper | Discussion Paper | Discussion Paper |

**TCD**

doi:10.5194/tc-2015-202

**Melt ponds impact ice concentration retrieval**

S. Kern et al.

ciated with ISF $\approx 65\,\%$ located closer to $TB_{MYI}$, yield SIC $\approx 70\,\%$ while TBs associated with the same ISF located at the FYI line or even to the right of it yield SIC $\geq 80\,\%$. Therefore, over-estimation of ISF by Bootstrap_p depends on ice type and is larger for FYI than MYI when using fixed winter tie-points.

5    TBs of a large range are associated with similar ISF values, either parallel to the Bootstrap_p ice line but at varying distance to $TB_{OW}$ or parallel to the FYI line. Bootstrap_p SIC over-estimates the ISF the more the more ISF decreases/MPF increases. This is indicated in plot (e) of Figs. 7 and 8 where a regression line, if it were not forced through (0,0), would have a slope $< 1$.

10    Worth mentioning is the considerable number of TBs associated with ISF $\approx 85\,\%$ even below – relative to the ice line – the location of the points with ISF $\approx 65\,\%$, i.e. a reversal of the ISF gradient mentioned above (Fig. 12a). These TBs are located at an estimated Bootstrap_p SIC between 60 and $75\,\%$ (see also Fig. 7e) and are an example of ISF under-estimation by Bootstrap_p. A more detailed look into Figs. 9 to 15 11 reveals that all algorithms except Bootstrap_f do have some data points of similar characteristics. Bootstrap_f is the only algorithm, which relies on TB at V polarization only. All other algorithms include also TB at H polarization.

Bootstrap_f and Bootstrap_p together form the Comiso Bootstrap algorithm (CBA) that is used for Arctic sea ice. Bootstrap_p is applied to TBs above the ice line (Fig. 12) 20 minus 5 K, which represents SIC $\geq 90\,\%$, while Bootstrap_f is applied to the rest of the data (Comiso et al., 1997). Therefore, when interpreting our results in the context of using the CBA for summer Arctic SIC retrieval one should always combine the results found for Bootstrap_f and Bootstrap_p.

Our estimates of summer TBs for sea ice at the frequencies used by the CBA in the 25 Arctic are shown in Table 3. We superposed them additionally onto Figs. 11 and 12 (cyan cross). For Bootstrap_f (Fig. 11) our summer ice tie point is located substantially above the winter ice line almost on the FY ice line. For Bootstrap_p (Fig. 12) our summer tie point is located below the winter ice line a substantial amount to the right of the FY ice line. Together with our summer tie points we provide the winter (black) and sum-

**TCD**

doi:10.5194/tc-2015-202

**Melt ponds impact ice concentration retrieval**

S. Kern et al.

Interactive Discussion

Discussion Paper | Discussion Paper | Discussion Paper | Discussion Paper |

mer (cyan) ice lines for CBA for SSM/I (Comiso et al., 1997). For Bootstrap_f (Fig. 11) the summer ice line is located above and almost parallel to the winter ice line. Both these lines are valid for SSM/I TBs and therefore have a slightly steeper slope than our ice line which is for AMSR-E; note the frequency difference of the 19 GHz chan-

nel used: SSM/I: 19.4 GHz, AMSR-E: 18.7 GHz. For Bootstrap_p (Fig. 12), Comiso et al. (1997) provide three different summer ice lines: for 1–18 July, for 19 July–4 August, and for 5 August–28 September. We plotted the first two as solid and broken cyan lines, respectively. While the first summer ice line has a substantially steeper slope than the winter ice line, the second summer ice line is located below the winter ice line and has

a slope quite similar to the winter ice line. Note that winter ice lines (black and thick red) are almost identical in location and slope.

From the location of our summer ice tie point and the summer ice line relative to the winter ice lines shown in Fig. 11 for Bootstrap_f it is likely that by using summer ice tie points and a summer ice line one would be able to reduce ISF over-estimation.

It can be assumed that a summer ice line for AMSR-E would be located above our winter ice line – perhaps in proximity to our summer ice tie point. A reduction of ISF over-estimation of up to 20 % would seem possible.

The summer ice lines given in Fig. 12 for Bootstrap_p are less straightforward to explain – mainly because of their different slopes. By applying the first (solid) line, ISF

over-estimation would increase on the MYI side while on the FYI side it would likely decrease. By applying the second (broken) line, one would systematically decrease the distance between $TB_{OW}$ and the ice line and would cause a larger ISF over-estimation than by applying the winter ice line. Our summer ice tie point is located on this second summer ice line. Note that open water tie points at 37 GHz change by only 0 and 1 K

at H- and V-polarization, respectively (Comiso et al., 1997).

Bootstrap_p is the part of the CBA which is used to decide which mode of the CBA is used for Arctic SIC retrieval. The location of the summer ice lines in Fig. 12 suggests that the switch between Bootstrap_p and Bootstrap_f mode, which is supposed to occur

Discussion Paper | Discussion Paper | Discussion Paper | Discussion Paper |

**TCD**

doi:10.5194/tc-2015-202

**Melt ponds impact ice concentration retrieval**

S. Kern et al.

at 90 % during winter conditions (see above), would occur at different SIC values – which according to our investigation would be substantially below 90 %.

We note that the summer ice line for Bootstrap_p for the third period: 5 August to 28 September (Comiso et al., 1997) (not shown in Fig. 12) is located again quite close to the winter ice line as slope and intercept are almost identical: 1.000 and −12.0 K for winter and 0.993 and −14.0 K for the third summer ice line, respectively.

## 4.6 Implications and concluding discussion

We would like to stress one more time that the parameters derived from the MODIS data (MPF and ISF) strongly depend on the quality of the cloud-masking scheme. Even though we restrict our analysis to cases with detected cloud cover < 5 % the actual cloud cover might have been higher. Because clouds have a higher reflectivity than open water, undetected clouds would cause a positive bias in ISF and a negative bias in MPF. A bias into the opposite direction would be caused by cloud shadows, which are particularly difficult to account for. We recommend considering an uncertainty in ISF and MPF of at least 5 to 10 %. As most of our results are based on July when melt has commenced everywhere, co-existence of melting and freezing conditions cannot be ruled out but are not so much of an issue.

In addition, only 3 different channels are used in the MPF retrieval algorithm (Rösel et al., 2012a) and it is therefore difficult to discriminate between more than three surface types (ISF, OWF, and MPF) even though this could improve the accuracy. A refined algorithm using more channels should operate with four surface classes by discriminating between melt ponds on MYI and FYI separately or even discriminate between bare and snow-covered sea ice, which would introduce a 5th surface class but in return reduce the uncertainty caused by merging different categories of snow and ice types.

At the frequencies used for SIC retrieval using satellite microwave radiometry, the radiometric contrast between FYI and MYI vanishes during summer (Eppler et al., 1992; Comiso and Kwok, 1996). This affects SIC retrieval under summer conditions. We found that the difference between SIC and ISF – basically an over-estimation of

Discussion Paper | Discussion Paper | Discussion Paper | Discussion Paper |

**TCD**

doi:10.5194/tc-2015-202

**Melt ponds impact ice concentration retrieval**

S. Kern et al.

Title Page

Abstract | Introduction

Conclusions | References

Tables | Figures

◁ | ▷

ISF in most of the cases – is a function of the ice type for Bootstrap_f, Bootstrap_p and NASA_Team for the data set investigated here. Our investigation confirms earlier studies in that we find the radiometric summer signature of (MY-) ice with an average age of 4 years or older to be more similar to the one of FYI, independent of frequency (19, 37, 89 GHz) and polarization (Figs. 9b to 12b).

Of the algorithms investigated Bootstrap_p seems to have the lowest sensitivity to surface property changes and provides SIC which agrees with ISF within ±10 % for ISF ≥ 90 % and hence MPF < 5 %. This is where Bootstrap_f shows the largest over-estimation of ISF in our results. Bootstrap_p, however, tends to over-estimate ISF more the more melt ponds cover the surface while it is the opposite for Bootstrap_f for which the estimated SIC shows a clear relationship to ISF (slope 1.4, correlation coefficient > 0.85): the smaller the ISF and hence the larger the MPF the smaller is the apparent over-estimation of ISF by Bootstrap_ f. ASI and NT2 algorithms were not included in this investigation due to their SIC cut-off at 100 %.

In July the sea ice can be expected to reveal the radiometric signature of melt or melt-refreeze. The snow can be expected to be moist if not already wet and saturated with melt water. The associated change in emissivity is expected to cause an increase in the TB compared to winter conditions. Depending on the geographic location and the progress of the melt season, also melt-refreeze cycles might still occur in July. Hence, radiometric properties can be expected to be quite variable, as is demonstrated by the TBs shown in Figs. 4 and 5, because moist and wet snow causing high TBs (e.g. Garrity, 1992; Cavalieri et al., 1990; Stiles and Ulaby, 1980) co-exist with coarse grained snow typical for re-frozen snow causing low TBs at the higher microwave frequencies (e.g. Lakhankar et al., 2013; Harouche and Barber, 2001; Crane and Anderson, 1994). These different surface and radiometric properties explain why in Fig. 5 the TB distribution for ISF > 95 % (grey curve) covers a relatively wide range, particularly at 37 and 89 GHz. Recent model investigations employing SNTHERM forced by ERA Interim reanalysis data underline the increase in the absolute values and in the variability of the microwave emissivity (at Special Sensor Microwave/Imager frequencies)

**TCD**

doi:10.5194/tc-2015-202

**Melt ponds impact ice concentration retrieval**

S. Kern et al.

of snow-covered sea ice towards early summer – especially at 85 GHz but also at 37 GHz (Willmes et al., 2014). Also the variability of the derived parameters PR19 and GR3719 is increasing substantially from May to June. Usually, once snow melt has commenced, snow melts out relatively quickly, i.e. within 1–2 weeks (Perovich et al.,
2014; Landy et al., 2014) leaving bare melting sea ice. This type of sea ice is – by itself – quite variable as well and causes, in response to different density, grain size, salinity and near-surface wetness, different penetration depths and different levels of volume scattering at the different frequencies involved in the present study.

We note that the change in SIC and in TBs with respect to ISF is – at least in the
10 data set considered here – largely based on MYI. In plot (a) of Figs. 10–12 only 120 data points belong to FYI while 2323 data points belong to (MY) ice of an average age of 4 years or older (see plot b) of the respective figures); 5484 data points are ice of an average age of 1 to 3 years. For NASA_Team, Bootstrap_f, and Bootstrap_p, a shift in the TB or TB parameters (GR3719V) from values near $TB_{MYI}$ towards values
near $TB_{FYI}$ is associated with a decrease in ISF and hence an increase in MPF (plot b of Figs. 10 to 12). This shift from a MYI to a FYI signature is illustrated nicely for NASA_Team in Fig. 6a) showing that the mode of GR3719V, which is a good measure for the MYI fraction, moves from a typical MYI value of $-0.1$ for ISF > 95 % to close to 0.0 for ISF > 75 %. For Bootstrap_f a similar behavior has been found in Comiso and
Kwok (1996).

## 5  Conclusions

Open water fraction (OWF) and melt pond fraction (MPF) obtained from daily quasi-clear sky reflectance data from the MODIS sensor (Rösel et al., 2012a) are inter-compared with co-located, contemporary AMSR-E brightness temperature (TB) data
and sea ice concentration (SIC) data computed from this TB data. The investigation focuses on the Arctic Ocean and the period June to August 2009. SIC is computed from the AMSR-E TB data using a set of eight different retrieval algorithms (Table 1)

**TCD**

doi:10.5194/tc-2015-202

**Melt ponds impact ice concentration retrieval**

S. Kern et al.

applying a consistent set of tie points (Ivanova et al., 2015). The investigation is carried out under the hypothesis that theoretically passive microwave sensors operating at, e.g., the AMSR-E frequencies cannot distinguish between open water between ice floes and open water in melt ponds on ice floes because penetration depth into water is a few millimeters at most. For this purpose we compute the ISF, which is 1 minus OWF minus MPF, and investigate how AMSR-E TB and SIC vary as a function of ISF. Our investigation focuses on the month of July because MPF and hence ISF changes are largest during July. In July 2009, 98.5 % of the grid cells used are predominately covered by multiyear ice (MYI), about 30 % of the grid cells used are covered by ice of an average age of 4 years or older. The majority of the grid cells selected as having a MODIS SIC > 90 % actually has SIC ≥ 95 %. This implies that variations in ISF between 60 and 100 % are mainly caused by variations in MPF, which varies between approximately 5 and 35 % in our study (Fig. 3). An uncertainty of 5 to 10 % in ISF and MPF should be taken into account.

We find that TBs and parameters derived from TBs such as the polarization ratio (PR) change non-linearly as a function of ISF (and hence MPF) during melt. For given ISF, TBs range over about 10 to 25 K at 6 and 19 GHz but may range over 50 to 70 K at 37 and 89 GHz. TB values do not change monotonically over the ISF range investigated – except at 6 GHz. For derived parameters such as TB polarization difference at 89 GHz and PR at 19 GHz the change is monotonic but the change per 10 % increase in ISF is substantially smaller than expected from theoretical computations assuming fixed ice and water signatures – independent of whether winter or summer values are employed.

It is noted that the observed TB changes are not solely caused by changes in ISF due to changes in MPF. The increasing variability of snow and ice radiometric properties during melt also contribute to these TB changes; these contributions are a function of frequency and polarization and have the potential to partly counter-balance the impact of changing MPF and ISF on the observed TB.

All investigated SIC retrieval algorithms over-estimate ISF when fixed winter tie points are applied. The amount of this over-estimation varies with ISF and hence MPF between the algorithms.

For algorithms where SIC retrieval is mainly based on the TB measured at one frequency but both polarizations (Table 1: Type P), ISF over-estimation is relatively small for high ISF values: 5 to 20 %. But for these algorithms, ISF over-estimation increases as ISF itself decreases: 10 to 40 %. Correlations between SIC and ISF are relatively small and the slope of a linear regression line forced through SIC = ISF = 0 % is between 1.1 and 1.2 but would be substantially < 1 if the line would be forced through SIC = ISF = 0 %. Type P algorithms are found to be more sensitive to the ice type: ISF over-estimation is larger for FYI or ice with a FYI-like radiometric signature than for MYI.

For algorithms where SIC retrieval is mainly based on the TB measured at two different frequencies but one polarization (Table 1: Type F), ISF over-estimation for high ISF can reach 40 %. However, in contrast to Type P algorithms, the over-estimation of ISF decreases as ISF itself decreases: 15 to 30 %. Correlations between SIC and ISF are ≥ 0.85 and the associated linear regression lines, which naturally intersect close to SIC = ISF = 0 %, and exhibit slopes of 1.3 or 1.4, suggest an exploitable relationship between SIC and ISF if reliable ice tie points can be established.

For one of the Type P and one of the Type F algorithms, Bootstrap_p and Bootstrap_f, we discussed the location in TB-space of winter and summer ice tie points as published for SSM/I (Comiso et al., 1997), our AMSR-E winter ice tie points (Ivanova et al., 2015), and an estimate of AMSR-E summer tie points based on our data. The three main results of this discussion are: (1) application of the SSM/I Bootstrap_f (our AMSR-E) summer ice tie points would slightly (substantially) reduce the over-estimation of ISF by SIC; (2) The locations of the three SSM/I Bootstrap_p summer ice tie points – or rather ice lines – in TB-space are quite different which would cause both over- and under-estimation of ISF by SIC; (3) application of the SSM/I Bootstrap_p summer ice tie point, which agrees most with our AMSR-E summer ice tie point, would substantially increase

**TCD**

doi:10.5194/tc-2015-202

**Melt ponds impact ice concentration retrieval**

S. Kern et al.

the over-estimation of ISF by SIC obtained with Bootstrap_p and would decrease the SIC threshold at which the Comiso Bootstrap algorithm switches between polarization mode and frequency mode to values considerably below 90 %.

As this study showed, melt ponds are interpreted as open water by the SIC algorithms, while the concentration of ice between the melt ponds is in general being overestimated. These two effects may cancel out each other and thus produce seemingly correct SIC for the wrong reasons. This cancelling effect will in general only be "correct" at one specific MPF. The fact that melt ponds are interpreted as open water has a physical explanation, and we recommend that the SIC obtained should not be corrected for this issue. However users should be aware that the SIC algorithms available at the moment retrieve a combined parameter presented by SIC in winter and ISF in summer. As for the overestimation of SIC, using a Type F algorithm, which allows biascorrection, together with a set of tie points retrieved dynamically could be a solution. Retrieving such a set of tie points is however a challenging task because we found TBs to change non-linearly with ISF and hence MPF and because other factors than the MPF increase may have a substantial influence on the TBs.

*Acknowledgements.* We thank the data providers: NASA DAAC for AMSR-E L2 brightness temperatures and MODIS L1B reflectance data, NSIDC for AMSR-E L3 sea ice concentrations. This work was funded by ESA/ESRIN (sea ice CCI). S. Kern acknowledges support from the Center of Excellence for Climate System Analysis and Prediction (CliSAP), University of Hamburg, Germany. A. Rösel, N. Ivanova, L. T. Pedersen, R. Saldo, and R. T. Tonboe acknowledge funding from the European Space Agency (ESA) Climate Change Initiative Sea Ice Project (SICCI).

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

**Table 1.** The sea ice concentration algorithms. See text for explanation 1 of column "Type".

| Algorithm | Acronym | Reference | Frequencies | Type |
|-----------|---------|-----------|-------------|------|
| Bootstrap_p | BP | Comiso (1986) | 37V, 37H | P |
| Bootstrap_f/ CalVal | BF | Comiso (1986) | 19V, 37V | F |
| Bristol | BR | Smith (1996) | 19V, 37V, 37H | M |
| NASA Team | NT | Cavalieri et al. (1984) | 19V, 19H, 37V | M |
| ASI | ASI | Kaleschke et al. (2001) | 85V, 85H | P |
| Near 90GHz linear | N90 | Ivanova et al. (2014) | 85V, 85H | P |
| One_channel (6H) | 6H | Pedersen (1994) | 6H | S |
| NASA Team 2 | NT2 | Markus and Cavalieri (2000) | 19V, 19H, 37V, 85V, 85H | M |

**TCD**

doi:10.5194/tc-2015-202

**Melt ponds impact ice concentration retrieval**

S. Kern et al.

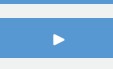

**Table 2.** Summary of values of slope, correlation, and RMSD from Figs. 7 and 8 for all 8 algorithms considered for July. For every algorithm values of the respective parameter derived from July only data for all grid cells come first, denoted by "All", followed by the values derived for MYI grid cells, denoted by "MYI". N90 denotes the algorithm Near90_lin and 6H denotes the algorithm One_channel 6H (see also Table 1).

| Algorithm | | Bootstrap_f | Bootstrap_p | Bristol | NASA_Team | 6H | N90 | ASI | NT2 |
|---|---|---|---|---|---|---|---|---|---|
| Slope | All | 1.42 | 1.13 | 1.32 | 1.22 | 1.33 | 1.23 | 1.29 | 1.28 |
| | MYI | 1.42 | 1.11 | 1.31 | 1.21 | 1.32 | 1.21 | 1.27 | 1.25 |
| Corr. | All | 0.85 | 0.43 | 0.85 | 0.66 | 0.81 | 0.53 | 0.42 | 0.38 |
| | MYI | 0.91 | 0.19 | 0.86 | 0.65 | 0.86 | 0.34 | 0.20 | 0.36 |
| RMSD | All | 33.3 | 15.3 | 25.1 | 19.2 | 26.1 | 20.8 | 25.3 | 24.7 |
| | MYI | 33.4 | 16.7 | 24.9 | 19.1 | 25.8 | 21.4 | 25.2 | 23.7 |

**TCD**

doi:10.5194/tc-2015-202

**Melt ponds impact ice concentration retrieval**

S. Kern et al.

**Table 3.** Top row: winter tie points for FYI and MYI expressed as polarization ratio PR; other rows: summer tie points derived as outlined in the text expressed as PR and TB at V- (TBV) and H-polarization (TBH). TBs are given together with one standard deviation.

| Frequency | 19 GHz | 37 GHz | 89 GHz |
|---|---|---|---|
| PR (winter, FYI; MYI) | 0.030; 0.043 | 0.025; 0.031 | 0.021; 0.024 |
| PR (summer) | 0.034 | 0.033 | 0.021 |
| TBH (summer) [K] | $247.6 \pm 6.5$ | $239.0 \pm 4.9$ | $226.3 \pm 10.0$ |
| TBV (summer) [K] | $265.2 \pm 2.5$ | $255.5 \pm 4.5$ | $235.0 \pm 11.8$ |

**TCD**

doi:10.5194/tc-2015-202

**Melt ponds impact ice concentration retrieval**

S. Kern et al.

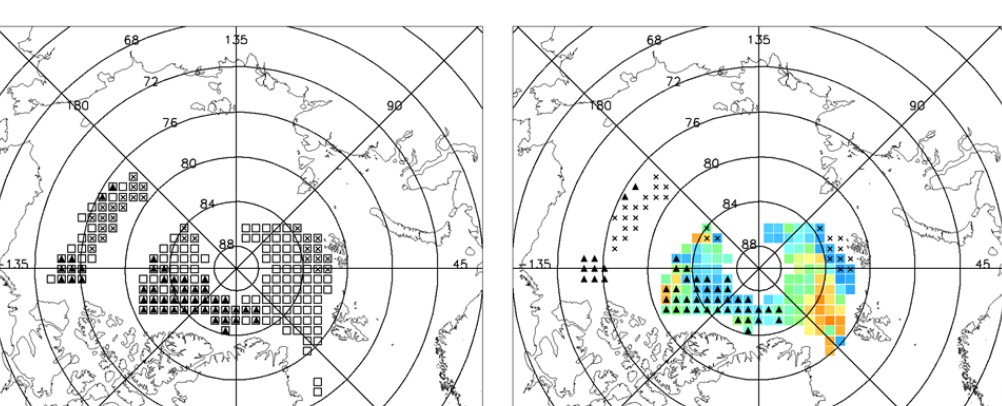

**Figure 1.** Location of the grid cells used in the present study. Left: all grid cells with MODIS melt pond data for June–August 2009. Right: MODIS melt pond fraction at the end of July 2009. In both images triangles and crosses denote multiyear ice (MYI) older – on average – than 4 years and first-year ice (FYI), respectively. Grid cells without symbols are covered by younger MYI. The right image exemplifies that in July no data are available from the Beaufort Sea south of 80° N and that only a very small fraction of usable grid cells is covered by FYI.

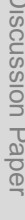

**TCD**

doi:10.5194/tc-2015-202

**Melt ponds impact ice concentration retrieval**

S. Kern et al.

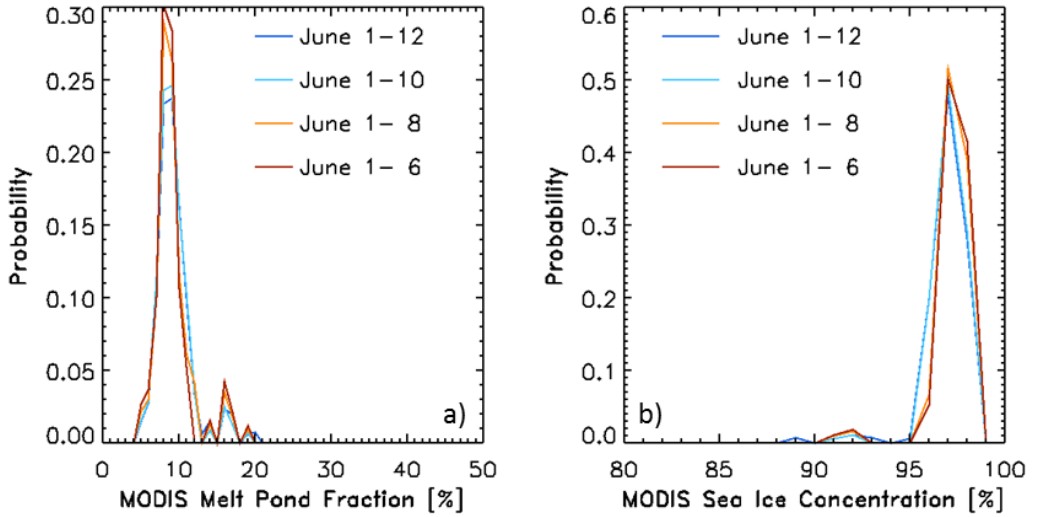

**Figure 2.** Histograms of MODIS melt pond fraction MPF **(a)** and MODIS sea ice concentration MSIC **(b)** derived when sea ice cover was near 100 % and melt ponds were not yet present (see text for details) for the first 7, 9, 11, and 13 days of June 2009.

TCD

doi:10.5194/tc-2015-202

**Melt ponds impact ice concentration retrieval**

S. Kern et al.

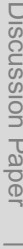

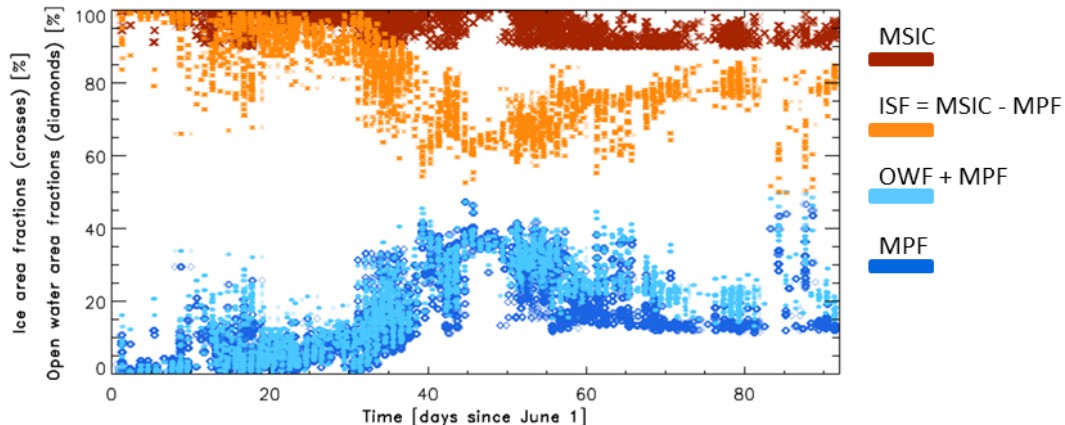

**Figure 3.** Time series of open water and sea ice fractions for all MODIS grid cells used in the present study for 1 June to 31 August 2009.

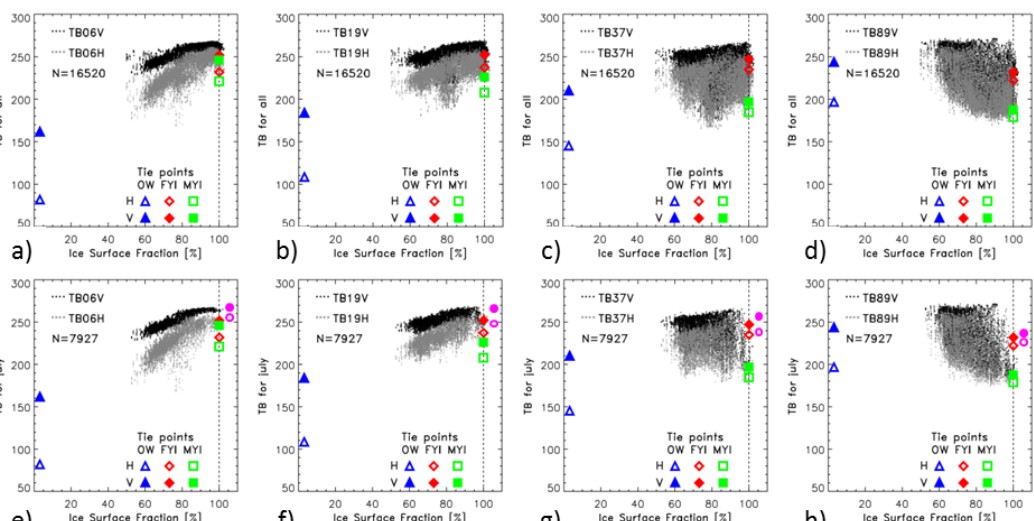

**Figure 4.** Brightness temperatures (TBs) observed by AMSR-E as function of the MODIS ice surface fraction. Top row contains all data; bottom row contains only data from July 2009. From left to right data are shown for frequencies: 6, 19, 37, and 89 GHz. Only data with a cloud fraction $< 5\%$ and MODIS SIC $> 90\%$ are shown. $N$ is the number of data points plotted. Note that symbols for horizontal (H) polarization are over-plotting those for vertical (V) polarization. Colored symbols denote winter tie points for open water (OW), first-year ice (FYI) and multiyear ice (MYI) taken from the Round Robin Data Package (RRDP) of the ESA-CCI sea ice ECV project (Ivanova et al., 2015) and, only in the bottom row, our summer sea ice tie points (purple), see Table 3 and text for details. Tie points for open water are shifted a bit to the right for better visibility.

**TCD**

doi:10.5194/tc-2015-202

**Melt ponds impact ice concentration retrieval**

S. Kern et al.

Discussion Paper | Discussion Paper | Discussion Paper | Discussion Paper

Discussion Paper | Discussion Paper | Discussion Paper | Discussion Paper

TCD

doi:10.5194/tc-2015-202

Melt ponds impact ice concentration retrieval

S. Kern et al.

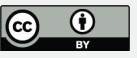

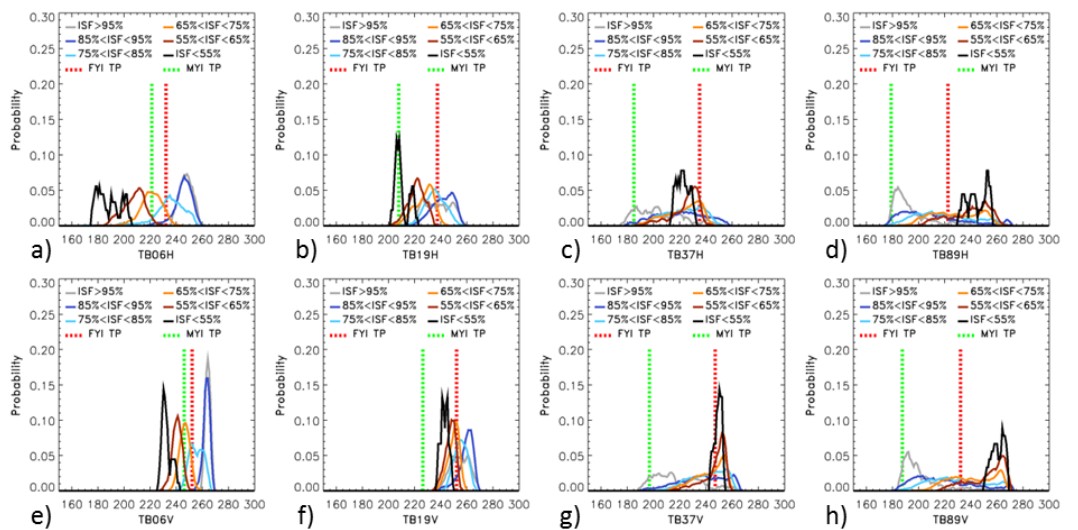

**Figure 5.** Normalized distribution of TB for different ISF bins for H (top) and V-polarization (bottom) for, from left to right, 6 GHz, 19 GHz, 37 GHz, and 89 GHz. Only data from July are used. Dashed vertical lines denote winter tie points for first-year ice (FYI) and multiyear ice (MYI) taken from the Round Robin Data Package (RRDP) of the ESA-CCI sea ice ECV project (Ivanova et al., 2015) (see Fig. 4). Our summer sea ice tie points (Table 3 and purple symbols in Fig. 4) are not included here; they would be located close to the FYI tie point lines.

Discussion Paper | Discussion Paper | Discussion Paper | Discussion Paper |

**TCD**

doi:10.5194/tc-2015-202

**Melt ponds impact ice concentration retrieval**

S. Kern et al.

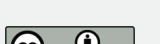

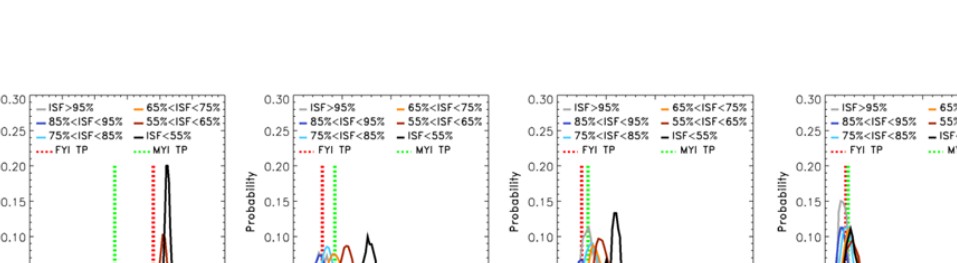

**Figure 6.** Normalized distribution of GR3719 **(a)** and PR at 19 GHz **(b)**, 37 GHz **(c)**, and 89 GHz **(d)** for different ISF bins (see Fig. 5). Only data from July are used.

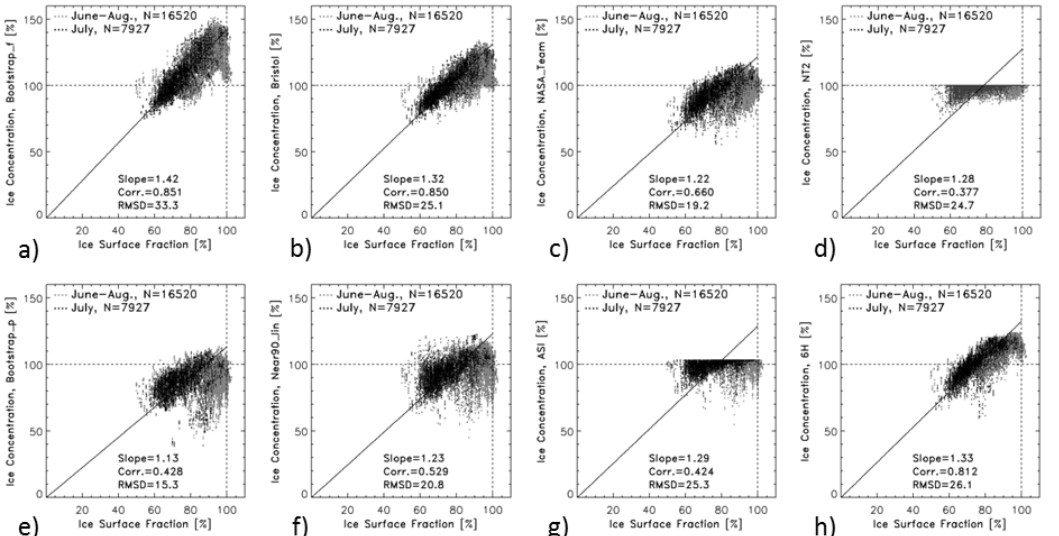

**Figure 7.** SIC derived with eight different algorithms (Table 1) from AMSR-E TBs versus ISF. Grey and black symbols denote data from June to August 2009 and July 2009 only, respectively. Note that black symbols are plotted on top and obscure grey symbols. *N* is the number of data pairs plotted. The solid line is the regression line computed from data points of July only, forced to intersect (0,0). The slope of the regression line is given together with the linear correlation coefficient (Corr) and the root mean squared difference (RMSD), both also for July, at the bottom of each image.

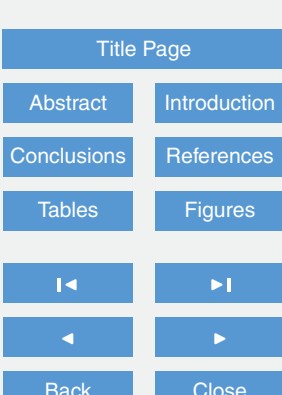

Discussion Paper | Discussion Paper | Discussion Paper | Discussion Paper |

**TCD**

doi:10.5194/tc-2015-202

**Melt ponds impact ice concentration retrieval**

S. Kern et al.

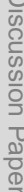

Discussion Paper | Discussion Paper | Discussion Paper | Discussion Paper

**TCD**

doi:10.5194/tc-2015-202

**Melt ponds impact ice concentration retrieval**

S. Kern et al.

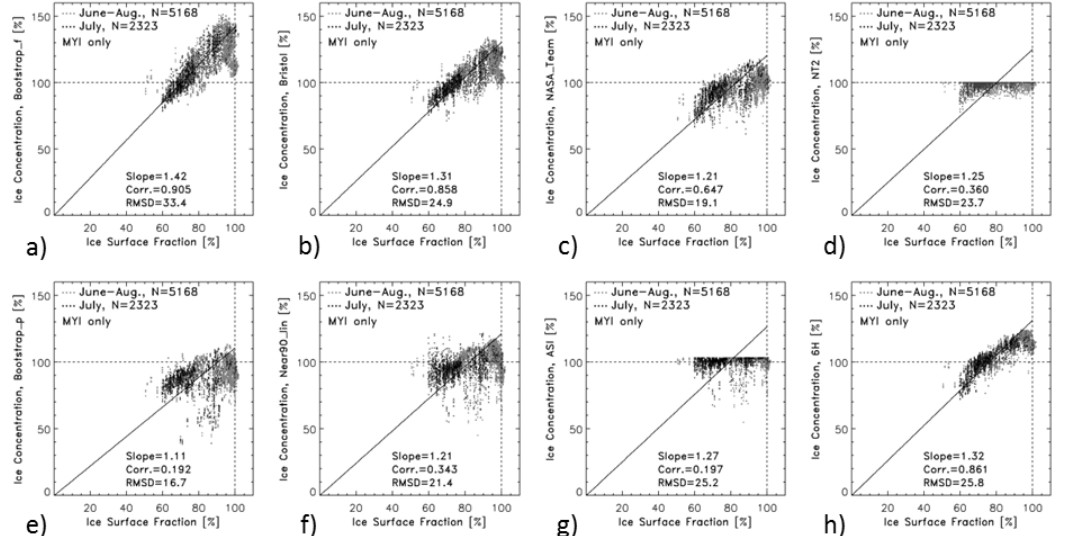

**Figure 8.** As Fig. 7 but only for grid cells defined as multiyear ice (see text at end of section "Data and methods").

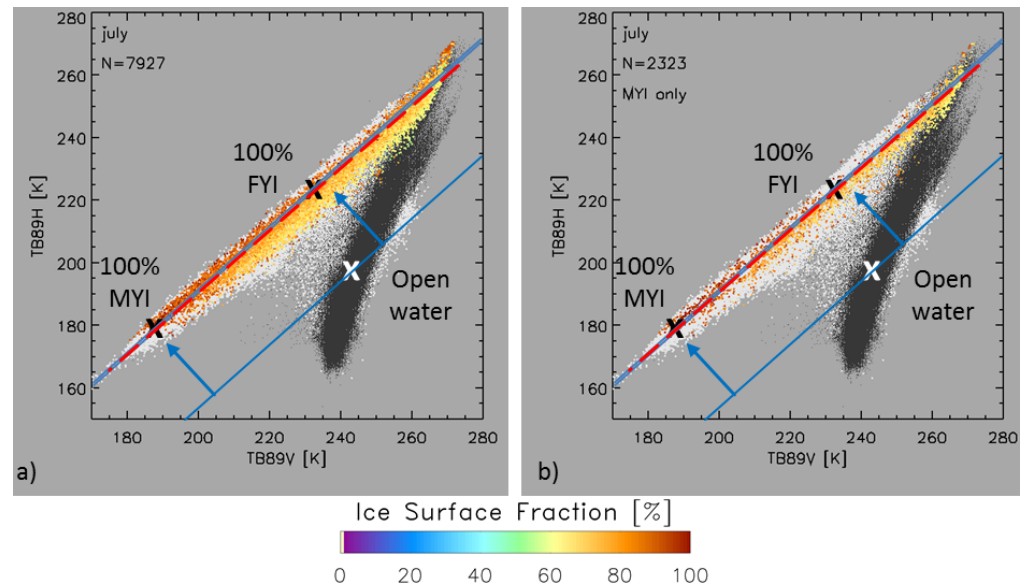

**TCD**

doi:10.5194/tc-2015-202

**Melt ponds impact ice concentration retrieval**

S. Kern et al.

**Figure 9.** Arctic Ocean daily AMSR-E TB at 89 GHz (TB89). Black and light grey symbols denote winter-time TB89 (10 February 2007) for NT2 SIC = 0 % and 1 % ≤ NT2 SIC ≤ 100 %, respectively. Where co-located the former over-plot the latter. Colored symbols denote TB89 for July 2009 with ISF from grid cells with < 5 % cloud cover, MSIC ≥ 90 % for all ice **(a)** and only MYI ice **(b)**. White and black crosses: winter tie points of open water, $TB_{OW}$, FYI, $TB_{FYI}$, and MYI, $TB_{MYI}$ (Ivanova et al., 2015) (see Fig. 4). The red dashed line approximates SIC = 100 %, assuming that SIC stays at 100 % between $TB_{FYI}$ and $TB_{MYI}$. Blue lines through $TB_{OW}$ and through $TB_{FYI}$ and $TB_{MYI}$ approximate lines of SIC = 0 % and SIC = 100 %, respectively. Blue arrows denote the direction of SIC increase. NT2 SIC is taken from the AMSR-E/Aqua Daily L3 12.5 km Brightness Temperature, Sea Ice Concentration, & Snow Depth Polar Grids product (http://nsidc.org/data/docs/daac/ae_si12_12km_tb_sea_ice_and_snow.gd.html, (Cavalieri et al., 2014)) available from NSIDC.

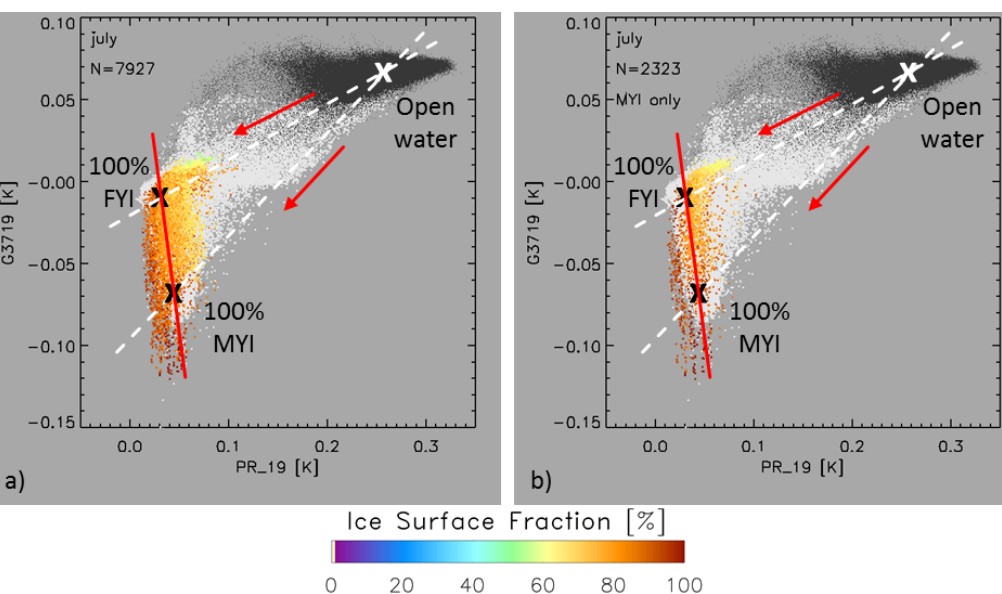

**Figure 10.** Arctic Ocean daily AMSR-E TB parameters GR3719 and PR19 representing the NASA_Team algorithm. Black and light grey symbols denote winter-time (10 February 2007) GR3719 and PR19 for NT2 SIC = 0 and $1\% \leq$ NT2 SIC $\leq 100\%$, respectively. Where co-located, the former over-plot the latter. Colored symbols denote GR3719 and PR19 for July 2009 with ISF from grid cells with $< 5\%$ cloud cover, MSIC $\geq 90\%$ for all ice **(a)** and only MYI ice **(b)**. For black and white crosses see caption of Fig. 9. The triangle formed by the red and the two white dashed lines approximates the NASA_Team algorithm tie point triangle (compare Cavalieri et al., 1990). Note that these are not straight lines in the original tie point triangle. The red line approximates SIC = 100% between $TB_{FYI}$ and $TB_{MYI}$ (ice line); the upper dashed line (FYI line) and the lower dashed line (MYI line) approximate the path along which SIC increases from 0% at $TB_{OW}$ to 100% at the ice line as indicated by the red arrows.

Discussion Paper | Discussion Paper | Discussion Paper | Discussion Paper |

**TCD**

doi:10.5194/tc-2015-202

**Melt ponds impact ice concentration retrieval**

S. Kern et al.

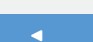 Title Page

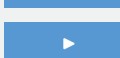 Abstract | Introduction

Conclusions | References

Tables | Figures

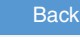 |◄ | ►|

Discussion Paper | Discussion Paper | Discussion Paper | Discussion Paper |

**TCD**

doi:10.5194/tc-2015-202

**Melt ponds impact ice concentration retrieval**

S. Kern et al.

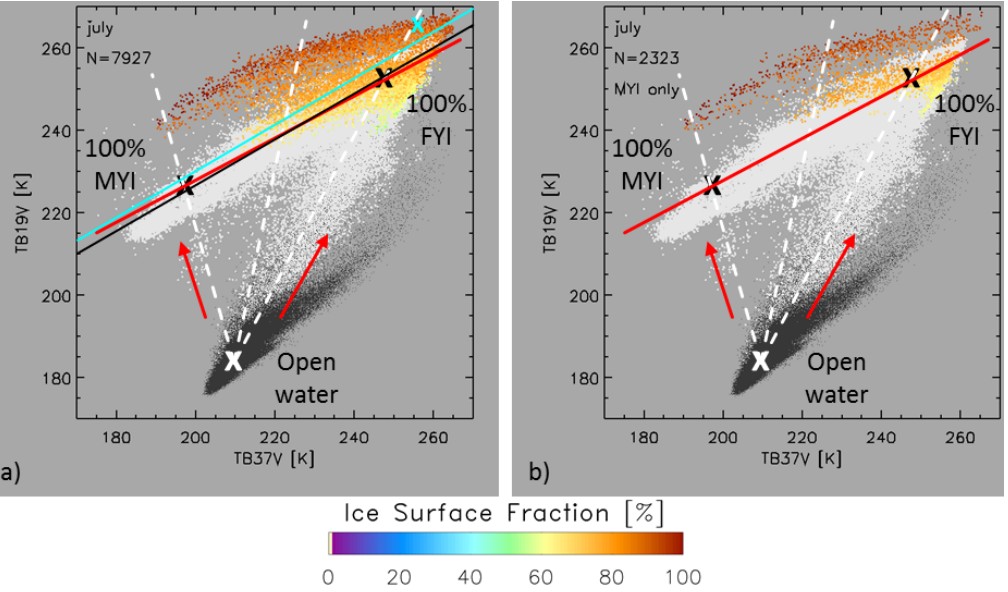

**Figure 11.** As Fig. 10 but for AMSR-E TBs at 37 and 19 GHz, V polarization, representing Bootstrap_f. The red line, leftmost and rightmost dashed white lines will be referred to as ice line, MYI line and FYI line, respectively, in the text. Thin black and cyan lines give the ice line of Bootstrap_f for SSM/I for winter and summer, respectively, according to Comiso et al. (1997). The cyan cross denotes our estimate of a summer sea ice tie point (see Table 3 and text for details).

TCD

doi:10.5194/tc-2015-202

**Melt ponds impact ice concentration retrieval**

S. Kern et al.

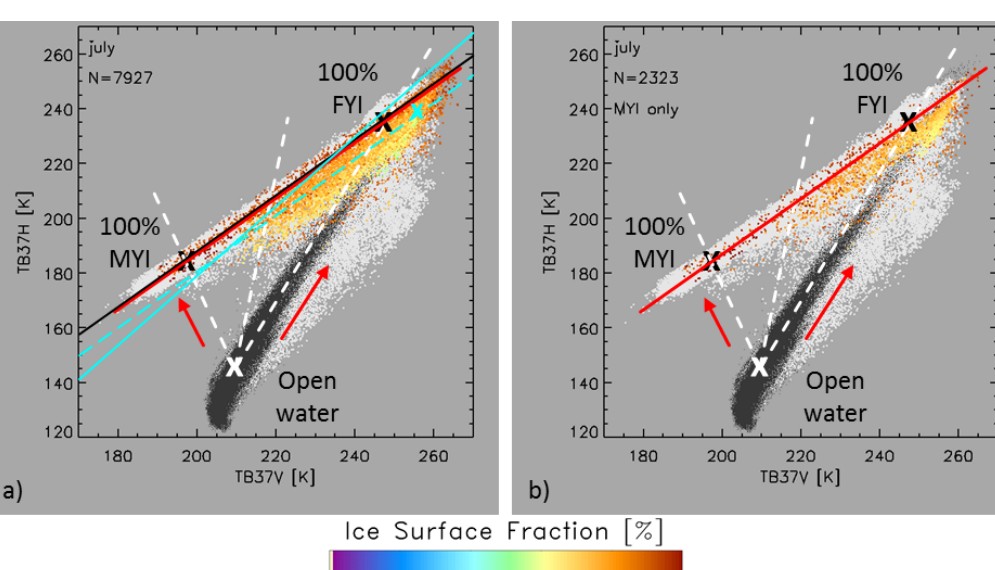

**Figure 12.** As Fig. 11 but for AMSR-E TBs at 37 GHz, V and H polarization, representing Bootstrap_p. Thin solid black, solid cyan and broken cyan lines give the ice line of Bootstrap_f for SSM/I for winter, summer: 1–18 July, and summer: 19 July–4 August, respectively, according to Comiso et al. (1997). The cyan cross denotes our estimate of a summer sea ice tie point (see Table 3 and text for details).

