# Peer review of "The impact of melt ponds on summertime microwave"

_The Cryosphere, 2015_

## Short Comment (SC1) · 25 Jan 2016

It was of interest to read the manuscript "The impact of melt ponds on summertime microwave brightness temperatures and sea ice concentrations" by S.Kern et al. The authors have conducted a very diligent and serious work. However, they state that from satellite microwave radiometer data it is impossible to distinguish open water between ice floes from open water in melt ponds on ice floes. I would refer the authors to a paper by Tikhonov et al. (2015). It presents a new algorithm called Variation Arctic/Antarctic Sea Ice Algorithm 2 (VASIA2) for determining sea ice concentration from satellite microwave radiometer data. VASIA2 not only retrieves ice concentration but also calculates ice areas covered by snow-water mixtures (SWM), including melted

snow and melt ponds.

Tikhonov V.V., I. A. Repina, M. D. Raev, E. A. Sharkov, V. V. Ivanov, D. A. Boyarskii, T. A. Alexeeva, N. Yu. Komarova. A physical algorithm to measure sea ice concentration from passive microwave remote sensing data. // Advances in Space Research. 2015. V. 56. N 8. P. 1578-1589. DOI: 10.1016/j.asr.2015.07.009.

Please also note the supplement to this comment:
http://www.the-cryosphere-discuss.net/tc-2015-202/tc-2015-202-SC1-supplement.pdf

**Supplement:**

[supplement omitted: unrelated document]

---

## Referee Comment (RC1) · Anonymous Referee #1 · 27 Feb 2016

In this paper Kern et al. discuss the potential impact of surface meltwater on passive microwave retrievals of sea ice concentration. The work is of interest to a broad community because it evaluates a well known, physics-based issue with sea ice concentration retrievals against an independent dataset. The methodology chosen is novel and appears to use the best available data type for comparison (MODIS melt pond retrievals from spectral mixing algorithms). The comparison is therefore useful and likely impactful. The data has issues however, and the potential exists for many other types of noise in the MODIS pond dataset to impact the conclusions the authors present. The authors present many caveats about drawing conclusions based on this comparison (e.g. Page 12 line 23). The reviewer felt the authors could do a better job clarifying

what conclusions remain firm and concrete regardless of all the errors and uncertainties, vs. which ones are on shakier ground. For example, it appears the conclusion that microwave models result in a value that exceeds ISF in summer is solid, but that the connection between ice type and overestimation is very sketchy (we develop this below). The reader could be more clearly provided this information.

The reviewer has significant reservations about the robustness of several of the conclusions reached, due to the substantial uncertainty in the MODIS melt ponds product used. These reservations are made much more pronounced because this work primarily uses gridcells above 85 degrees latitude, where MODIS, which is in a sun synchronous orbit, does not directly overpass. The off-nadir observations of surface reflectance at these high latitudes incorporate higher atmospheric path length and are much more impacted by surface roughness (the high parts of a rough surface are over-represented). Observations in this area are also impacted by low solar zenith angle, resulting in considerable shadowing, which has significantly different impacts on ascending and descending passes of MODIS, due to viewing geometry. Impacts of clouds and particularly cloud shadowing are higher at large off nadir angles as well. Selecting only ice of high concentration also necessarily subsets the MODIS data. The quality of this subset of the MODIS data should be addressed. There are many reasons why the data would not be as robust within a single extreme of the spectral mixing solution, such as near 100% concentration. For example, the spectral space between water and ponds is relatively small, and ocean water (even where not underlain by ice) at the edge of floes has a blue spectral signature more similar to ponds. This is due to both atmospheric distortion and scattering light transmitted through the ice in the upper ocean adjacent to floes. What is the potential that such narrow leads are interpreted as melt pond fraction more commonly in high ice concentration – leading to an import 'noise' in the MODIS pond data that is unique to high ice concentrations?

The reviewer feels the authors have not adequately addressed whether the passive microwave algorithms perform as designed – a critical question for most readers. The

key here is that the algorithms ARE NOT designed to produce ISF in summer, even if theoretically that is what they SHOULD see. These products are SIC products. So over-predicting ISF actually may indicate that the products are behaving exactly as designed – and are therefore empirically adapted to overcome the fact that the response should be based on ISF. The reviewer feels the authors must plot MSIC against SIC and evaluate whether the algorithm is actually working for the wrong reasons, rather than stating that the algorithm SHOULD theoretically produce ISF, and because it over-predicts this, it is inaccurate. Discussion and plots of ISF vs SIC can be retained and discussed in great detail, but the paper should not be published without comparison of SIC and MSIC.

The reviewer believes the author's work would have more impact in the future if more time is spent on refining and clarifying its presentation. As an overall impression, the reviewer found the results section quite dense and noted considerable redundancy in some discussion throughout the paper. The reviewer suggests authors try to consolidate statements to their appropriate sections to reduce these redundancies. The reviewer also noted excessive detail in describing some sections of the results that was not paired with relevant analysis – this made some sections a bit rote. Some of these descriptions of the data could be reduced and organized following clear statements about what they show. The reviewer also strongly encourages having an editor go over the text. There are many punctuation errors (particularly dozens of missing commas) and many instances of plural subjects with singular verb conjugations (i.e. the sentence on page 10, line 22-23) and several other odd wordings which may be hard for a non-native English speaker to eliminate.

There are lots of acryonyms being made up in this work (TB, ISF, SIC etc). The casual reader will not read the paper from end to end and/or may have different ideas of what these mean from prior works. As written, a thorough reading is required to find and becoming conversant in all these new acronyms. The reviewer strongly suggests a table of acronyms be created and placed into the document near the beginning. The

reviewer also regularly became confused about the origin of particular data products. Since the key to the entire paper is a comparison of MODIS-derived vs. microwave-derived products, all MODIS derived products should be somehow clearly differentiated from all AMSR-E or SSM/I derived products in the acronyms (ISF, for example is MODIS-derived, but not denoted as such in a manner similar to MSIC). Perhaps all MODIS derived product acronyms would start with 'm'.

Specific comments 1. Several times it is mentioned that brightness changes in the sea ice surface itself may counteract some of the melt pond covering. Please quantify the relative magnitude of brightness changes compared to melt pond flooding. 2. Page 2 line 18. These references are not the most appropriate for describing the physical processes of melt pond formation. Eicken et al., 2004; Polashenski et al., 2012; and Landy et al., 2014 are more focused on physical processes of pond formation. Perovich and Polashenski, 2012 is primarily focused on the evolution of albedo, as is Perovich et al., 2003. Petrich does discuss the connection between snow and pond locations. 3. Page 2 line 22 – "can cover up to 50-90%" this is an un-necessarily sensationalist statement, particularly for a paper which is trying to quantify the TYPICAL impact of ponds on SIC retrieval. It would be more appropriate to discuss the TYPICAL coverage of melt ponds rather than the EXTREME bound. The references here are also not particularly relevant. Eicken is not primarily focused on pond coverage but rather on the processes controlling ponding, papers published by Perovich in 2011 only reference other direct works on melt ponds. Yackel and Barber is appropriate, but only one of many. Also, Landy et al., 2014; Polashenski et al., 2012; Hanesiak and Barber etc. Futher, some of the references here and elsewhere are not found in the reference list (e.g. Perovich et al., 2011) 4. Page 2 line 25 – Albedo values certainly vary, but these albedo values are simply incorrect. Dry snow covered ice has an albedo of about 0.8. Bare, unponded melting FYI has a value of about 0.55 +- 0.1, depending on reference, and MY has 0.6+- 0.1. Melt pond covered ice tends to be lower than 0.5. Ponding therefore does not reduce ice albedo from 0.8 to 0.5, but rather from somewhere in the range 0.45-0.7 to somewhere in the range 0.1 to 0.5. Perovich, 2003 is a good

reference for MYI, but does not reflect current state of the literature which increasingly works on FYI. Perovich and Polashenski, 2012 describes FYI ponds, as does Frey et al., 2014. 5. Page 3, line 7 – This discussion of noise is important as the justification for trying to tease out why SIC from passive microwave products might be 'right for the wrong reasons'. It is worded poorly, and hard to follow. The project name and reason for conducting it is also not so important and could be dropped. The key is that the paper is largely about understanding whether the passive microwave products are interpreting changes in pond coverage as a type of noise in the SIC record. 6. Page 3 line 15 "Melt ponds are pools. . . " Redundant. this was already established and could be deleted. 7. Page 3 Line 16 - on penetration depth of passive microwaves. Passive microwaves are emitted from the sea ice and snow – as is correctly stated here. They are attenuated by liquid water. Discussing them as if the 'penetration depth' is limited would be language more appropriate to an active sensor with energetic waves PENETRATING from above. In this case waves EMITTED from below are being attenuated along the path to the sensor (because it goes through water). A novice reader could better understand that the ponds are attenuating a signal from below. 8. Page 5 MODIS data – Pond algorithm is executed using C5 data – several recent papers have suggested that significant uncorrected sensor degradation was present on C5 data (e.g Lyapustin et al., 2014). Though the degradation is only a few percent, it is not the same on all bands, meaning spectrally based algorithms can be significantly impacted. It would be useful to comment on whether this impacts the MODIS melt pond retrieval meaningfully. 9. MODIS data- It appears the locations you are focusing on, with high ice concentration, are all very far north. Here the MODIS data quality is likely to be quite poor because MODIS sun synchronous orbit does not place the sensor over high northern latitudes, and all retrievals are made at high off nadir angles, with low solar zenith angles, through long atmospheric path lengths. Under these conditions the MODIS surface reflectance products are well known to have substantial issues. The authors must at address this. The reviewer feels this is an important enough issue that the authors must examine whether the conclusions about SIC over representation by

passive microwave method apply at lower latitudes where MODIS data is likely better. Such a comparison may reveal that the FYI /MYI differences are not actually cause by ice type. This may require restricting the timeframe evaluated if 100% ice coverage is required. 10. Page 6 Paragraph 3, bias correction – Where does the 3% global addition to the MSIC come from? 11. Page 8 paragraph 3 – Ice age. It should be discussed and understood that the ice age within a 4 year cell is actually mostly less than 4 years, because all leads forming over the 4 year duration that at least some ice remained in the area refroze as younger ice. Many 4+year packs are composed of a large fraction of younger ice. As a result, this may not be a very effective mechanism for avoiding the influence of FYI. If the authors wish to really focus on the MYI/FYI differences they may consider using back trajectories to examine where the MYI was at the end of summer in the previous year, and eliminating MYI originating from areas of low ice concentration (where likely much FYI formed between the MYI floes). 12. Page 9 line 14. Morassutti and LeDrew primarily show that depth of the melt pond is not causally related to spectral response in visible wavelengths, but rather related to the underlying ice properties. . . so the reference should be after the first clause, and the second should be deleted. Deeper ponds on MYI actually appear spectrally similar to shallow, early season ponds on FYI- again because the predominant factor is underlying ice properties. 13. Uncertainties of MODIS sets – this section would benefit from a summary/concluding statement. In total, adding up all these errors in the MODIS sets, do they or do they not have the potential to alter the fundamental conclusions of this paper. 14. Line 23-25. This sentence would be better stated "Passive microwave emissions from the sea ice are attenuated within a path length of several mm of water, at the frequencies. . . hence in theory, melt ponds fully attenuate microwave emissions. . . and appear the same as leads.' Discussion of penetration again may confuse the novice reader into thinking this is some type of active sensing. Same comment applies to top of page 15 in section 4.1 15. Page 11 line 11 – these other factors impacting brightness temperature are very important and they are brought up repeatedly but never really addressed. This redundancy should be eliminated and a more complete discussion of

them should be included. What is the range of expected impacts from surface wetting on TB? For example- can the authors show that these are secondary in magnitude to the pond impacts? The reviewer is left concerned that the increasing brightness temperature of wet and metamorphosing snow could offset pond reductions in TB – leaving SIC algorithms right for the wrong reasons. 16. Page 12 last paragraph and page 13 first paragraph. Here would be a good place to quantify the TB change associated with these other changes. 17. Section 4. This section is dense and challenging to get through for all but the most intrepid reader. Reviewer suggests it will have more impact if presented more concisely, perhaps with several tables displaying results as a matrix of algorithm with over/under prediction. 18. Why not plot MSIC against SIC – perhaps the algorithm is actually working for the wrong reasons. 19. Page 17 Line 1- what gives the authors confidence that the range of bias in ISF does not exceed 10%, particularly given the small subset of the MODIS Melt pond data and extreme northern latitudes investigated? The reviewer is not convinced of this level of accuracy. 20. Page 19 Paragraph 2. This discussion about other factors strongly argues that this analysis should include investigation of smaller areas separately, so that impacts of melt timing can be considered. Last sentence of this paragraph is very long. The reviewer feels that such an analysis could greatly strengthen this work. 21. Section 4.3 Paragraph 1 and 2. This would seem to indicate that the NASA Team algorithm is behaving CORRECTLY at its stated purpose – observing SIC. Perhaps for the wrong reasons, but nevertheless, the NASA team algorithm does not target ISF as the authors reason it should, it targets SIC. It is not accurate, therefore to state that ISF is overestimated by NASA_Team algorithm, because this is not what the algorithm purports to produce. A note of this must be made here. Also, how does the team algorithm work? Does it effectively include an empirical correction that is handling a presumed melt pond fraction? 22. Section 4.4. This section is un-enlightening. The authors describe the plots but fail to discuss what these results mean or why they occur. 23. Page 22 line 28 – yes, but what would this reduction do to actual SIC – the parameter that the algorithm is designed to retrieve. 24. Page 23 line 21 – this reviewer is not convinced that the

MODIS ISF and MPF have this low of a bias under the extreme circumstances of high latitude, high off nadir angle, low solar zenith angle in the study region. 25. Page 23 line 24 – The reviewer finds this statement to be theoretically accurate but poorly informed. While the addition of more channels could theoretically increase the number of surface types discriminated, the spectral signature of FY ponds + MY ponds as well as Bare and Snow covered sea ice overlap considerably. Additional channels are extremely unlikely to add orthogonal information in this system and further differentiation is unlikely. 26. Page 24 line 3 – this reviewer is not convinced that the ice type relationship comes entirely from the microwave side of the data. MODIS ISF is also likely to be impacted by ice type, both due to changes in the spectral character of MY ponds vs FY ponds and due to the MY ponds commonly being more deeply recessed into the ice surface, and therefore less visible at high of nadir angles of MODIS. Also, see next comment. 27. Page 25 Line 29 – this is a very large majority of the data concentrated in MYI. This weakens the conclusions based on ice type considerably. Further, all the FYI is at a lower latitude, where the geometry of the satellite sensors is quite different. The reviewer feels that view geometry must be eliminated as a cause for the FYI MYI discrepancies if the authors are to retain discussion of FYI and MYI 28. Page 26 Line 11-15 – this is approximately the 5th time these other factors are mentioned in the paper. A considerable redundancy. Further, the reviewer finds none of the discussions of these other factors sufficiently quantitative for the reader to assess whether they are so large in magnitude as to alter the paper's fundamental conclusions. 29. Page 27 Line 14 – this statement is the thesis of the paper. The reviewer does not believe it has been adequately supported yet, because the authors seem to be ignoring that the SIC retrieval algorithms are calibrated for SIC retrieval, NOT ISF – even though, theoretically, ISF is the response they see. Melt ponds SHOULD be interpreted as open water based on theory. The data however, actually does not indicate that the algorithms ARE interpreting the ponds as open water. The over estimation of ISF means that the value produced is actually closer to SIC. This would mean that the algorithms are NOT interpreting the ponds as open water. Further, the statement below noting that the current

SIC algorithms produce SIC in winter and ISF in summer is also theoretically true but in practice unsupported by the data presented. The values do not represent ISF well. They are too high. They appear likely to represent SIC better. (Though such a comparison needs to be made) The reviewer believes the authors actually understand this distinction, but does not feel this has been clearly communicated to the reader yet. 30. Page 26 lines 27-33 – Further comparison is needed here between MSIC and SIC. Perhaps the algorithm has empirically corrected for MPF impact, for all the wrong reasons. An empirical algorithm which did this would likely produce above-100% values for 100% ice cover, non ponded. (ISF =100) Since these would be truncated to 100% by the user, this would not result in a SIC error in practice. Over estimation could also be correct if this were a case of 100% ice cover with 30% ponds, because the algorithms are supposed to be retrieving SIC. In this case it appears likely that MODIS would retrieve an ISF =70% while several of the algorithms would find SIC = 100%. Again both could be correct at their design function, even if theory says the microwave derived SIC should be seeing something else.

---

## Short Comment (SC2) · 9 Mar 2016

The ability to remotely sense and discriminate between sea-ice, open water and melt ponds is an important topic for model evaluation, process studies and initialization of operation forecast systems. This paper provides a very in-depth development and application of a scheme to estimate aspects of the area fraction of the surface types listed above.

Unfortunately, the paper is very difficult to read. It is so densely packed with abbreviations and acronyms and excess technical detail that it will be unintelligible to most readers. Even if one has a background in this field, the presentation style is a real impediment to effective communication. I would very strongly urge a complete re-write

of the manuscript with an eye to simplification, clarification and more organized flow of material. The same issues arise with the figures which are, like the text, almost impenetrable. I am convinced that the length of the text and the number of figures could be reduced by half, which would also allow individual figure panels to be increased to a readable size. This requires careful consideration of what the key messages/conclusions are, and what is the essential material that must be presented in order to substantiate these. There is no point publishing a paper that no one can or will want to read.

I read and re-read the Conclusions section multiple times and I must say I am still not clear on what the real take-home message is. Certainly there is a lot of detail about uncertainties and their source and the differences between different algorithms. But the last paragraph basically just says melt pond fraction is confounded with open water fraction in summer (something that has been well known since the early days of sea-ice remote sensing), that users should be aware of this, and that there nothing at the moment that can be done about it. Given all the preceding detail, it is surprising that nothing is said regarding which algorithms are more or less reliable and how a user might make choices when faced with a particular problem or application, or indeed how an 'essential climate variable' might be constructed. The second-last paragraph of the paper seems to provide some commentary on different algorithms, but having read it several times, I still cannot glean any concrete guidance.

So, my conclusion is that this paper requires quite a bit of work, and I would recommend major revisions.

---

## Author Comment (AC3) · 6 Jun 2016

The comment was uploaded in the form of a supplement:
http://www.the-cryosphere-discuss.net/tc-2015-202/tc-2015-202-AC3-supplement.pdf
* * *

---

## Author Response (AR1)

Dear editors, dear reviewers,

Thank you very much for giving us the opportunity to revise our manuscript. Because we got two major reviews, one of which was suggesting a complete rewrite, we only give a point-by-point reply where possible without copy-and-pasting pages from the revised manuscript into the reply to the reviewers' comments. We therefore ask the reviewers to please see the reply to the comments together with the completely re-written manuscript.

**Anonymous Referee #1**

In this paper Kern et al. discuss the potential impact of surface meltwater on pas- sive microwave retrievals of sea ice concentration. The work is of interest to a broad community because it evaluates a well known, physics-based issue with sea ice con- centration retrievals against an independent dataset. The methodology chosen is novel and appears to use the best available data type for comparison (MODIS melt pond re- trievals from spectral mixing algorithms). The comparison is therefore useful and likely impactful. The data has issues however, and the potential exists for many other types of noise in the MODIS pond dataset to impact the conclusions the authors present. The authors present many caveats about drawing conclusions based on this comparison (e.g. Page 12 line 23). The reviewer felt the authors could do a better job clarifying what conclusions remain firm and concrete regardless of all the errors and uncertain- ties, vs. which ones are on shakier ground. For example, it appears the conclusion that microwave models result in a value that exceeds ISF in summer is solid, but that the connection between ice type and overestimation is very sketchy (we develop this below). The reader could be more clearly provided this information.

We thank the reviewer for the very careful review of the manuscript and the many very useful comments which we tried to accommodate where possible and for which we give a point by point response.

In the revised manuscript, which has been rewritten completely, we are clearer about the quality of the MODIS sea-ice parameters. Even though we also revised the discrimination into ice types we refrained from putting too much weight on these now. We still keep statistical parameters of the comparison between AMSR-E sea-ice concentration and MODIS sea-ice parameters in the tables for reference. We also kept the information about ice types in the algorithm parameters spaces as these help to describe what happens to the brightness-temperature data during summer.

**The reviewer has significant reservations about the robustness of several of the con- clusions reached, due to the substantial uncertainty in the MODIS melt ponds prod- uct used. These reservations are made much more pronounced because this work primarily uses gridcells above 85 degrees latitude, where MODIS, which is in a sun synchronous orbit, does not directly overpass. The off-nadir observations of surface reflectance at these high latitudes incorporate higher atmospheric path length and are much more impacted by surface roughness (the high parts of a rough surface are over-represented). Observations in this area are also impacted by low solar zenith an- gle, resulting in considerable shadowing, which has significantly different impacts on ascending and descending passes of MODIS, due to viewing geometry.**

The authors are well aware of the quality of the MODIS products in high latitudes with all its consequences. The MODIS melt pond product has to deal with shadowing of ridges and clouds, longer atmospheric pathways. The weekly MODIS product MOD09A1 is using per definition of the "MODIS Surface Reflectance User's Guide" only the pixel with the highest quality flag score. The quality flags are as follows:

1 BAD data derived from a faulty or poorly corrected L1B pixel
2 HIGHVIEW data with a high view angle (60 degrees or more)
3 LOWSUN data with a high solar zenith angle (85 degrees or more)
4 CLOUDY data flagged as cloudy
5 SHADOW data flagged as containing cloud shadow
6 UNCORRECTED data flagged as uncorrected
7 CLIMAEROSOL data flagged as containing the default level of aerosols
8 HIGHAEROSOL data flagged as containing the highest level of aerosols
9 GOOD data which meets none of the above criteria

This shows that a "pre-selection" of the pixels was done. We included a note about this in the revised manuscript.

To add to the point 2: On http://cloudsgate2.larc.nasa.gov/cgi-bin/predict/predict.cgi one can predict MODIS orbits together with all the local angles. At 86.5N, for instance, one gets 5 overpasses with a sensor viewing zenith angle < 50°, and 8 with a respective angle < 60°. The minimum angle is around 40° for 2-3 overpasses / day. At 87.5 °N we still get 4 overpasses / day with a sensor viewing zenith angle < 50° and about two overpasses are close to 45°. And this is actually the northern limit up to which MODIS sea-ice parameters are computed. This latitude limit is visible in Figure 1. We also encourage the reviewer to take a look at the 8-day MODIS melt pond fraction product under: http://icdc.zmaw.de/1/daten/cryosphere/arctic-meltponds.html to gain more confidence in this data set.

**Impacts of clouds and particularly cloud shadowing are higher at large off nadir angles as well.**

In addition to this, the authors reduce the influence of clouds and cloud shadows by using only 100 km grid cells with a total cloud cover < 5%. We investigated whether the results are sensitive to using a 10%, a 5% or a 2% cloud cover threshold (see the following figure).

[Figure]

Comparison of the impact of clouds: If we would use only data where the cloud cover
is < 2% we would obtain the images on the left; if we would use < 10% we would obtain
the images on the right. As we can see in this example, slope, correlation and RMSD
remain almost constant even though N varies by about 20 % between each of the cloud
fractions used. We think that using 5% as a threshold is a reasonable thing to do.

**Selecting only ice of high concentration also necessarily subsets the MODIS
data.**

The authors agree. However, two reasons speak for this sub-setting. One is that
we are particularly after looking for potential biases at high sea-ice
concentrations. It is known that sea-ice concentration algorithms have larger
uncertainties over lower sea-ice concentrations anyways. But at high sea-ice
concentrations an accuracy better than 5% is achieved in terms of both, accuracy
and precision, for winter. We aim to look at the summer situation – which
requires to subset the MODIS sea-ice parameter data set. The second reason is,
that indeed if we would use 50% as a threshold for the MODIS sea-ice
concentration, then the results would not change too much either because the
fraction of sea-ice concentrations < 90% is quite small in the MODIS sea-ice
parameter data set used. Apart from that, by using only 50% MODIS sea-ice
concentration as a threshold we would not be able to isolate the influence of the
melt ponds. We illustrate the latter in the following three figures. Based on these
sample figures and the other results (not shown) we feel confident that the sub-
setting to MODIS sea-ice concentrations > 90% is a reasonable thing to do.
Nevertheless we point out that it will be desirable to limit the investigation to >
98% because this gives a better correlation with similar slopes of the linear
relationship between AMSR-E sea-ice concentration and MODIS ice-surface
fraction.

[Figure]

Do we gain something by not using a MODIS sea ice concentration threshold of 90 % (left) but instead use, for instance a threshold of 50 % (right)?

→ In these sample plots for Bootstrap_f (top) and NASA-Team (bottom): no, we don't get anything new.

[Figure]

In these plots which show only the multiyear ice case (left 90%, right 50%): → no, the main message for the two algorithms shown is the same, slope, correlation and RMSD remain almost unchanged.

[Figure]

In these plots which show the first-year ice cases (left 90%, right 50%): → no, again slope and RMSD remain almost constant, only correlations improve because we include more data.

The quality of this subset of the MODIS data should be addressed. There are many reasons why the data would not be as robust within a single extreme of the spectral mixing solution, such as near 100% concentration. For example, the spectral space between water and ponds is relatively small, and ocean water (even where not underlain by ice) at the edge of floes has a blue spectral signature more similar to ponds. This is due to both atmospheric distortion and scattering light transmitted through the ice in the upper ocean adjacent to floes. What is the potential that such narrow leads are interpreted as melt pond fraction more commonly in high ice concentration – leading to an import 'noise' in the MODIS pond data that is unique to high ice concentrations?

The MODIS melt pond product was validated with observational data (although only very little), but the validation studies showed reasonable results. The effect of "mixed" pixels within the ice pack caused by edges of leads or lateral flooding of ice floes was already discussed in the original paper of Rösel et al., 2012.

We agree with the reviewer that the spectral space between water in leads and melt ponds is relatively small and that is could be that melt ponds are interpreted as open water (=leads) or open water (=lead) are interpreted as melt ponds. We would like to note though, that our main results root on the net sea-ice surface fraction for which the above-mentioned misclassification does not matter too much. The spectral space between melt ponds and the water surfaces (be it leads or melt ponds) is larger. If our investigation would focus on MODIS sea-ice concentrations only, then we would need to take this issue more into account than we actually do.

We don't think that narrow leads are particularly important for misclassifications because – like melt ponds – these are sub-grid scale phenomena and contribute similarly to the classification result.

Finally, we would like to add that the temporal development of the spectral signal of the melt ponds could cause the above-mentioned misclassification to occur more often later in the melt season because once melt ponds on first-year ice have deepened their spectral signature is closer to the one in leads. In other words, we expect such misclassifications to be more relevant in August than in June.

**The reviewer feels the authors have not adequately addressed whether the passive microwave algorithms perform as designed – a critical question for most readers. The key here is that the algorithms ARE NOT designed to produce ISF in summer, even if theoretically that is what they SHOULD see. These products are SIC products. So over-predicting ISF actually may indicate that the products are behaving exactly as de- signed – and are therefore empirically adapted to overcome the fact that the response should be based on ISF. The reviewer feels the authors must plot MSIC against SIC and evaluate whether the algorithm is actually working for the wrong reasons, rather than stating that the algorithm SHOULD theoretically produce ISF, and because it over- predicts this, it is inaccurate. Discussion and plots of ISF vs SIC can be retained and discussed in great detail, but the paper should not be published without comparison of SIC and MSIC.**

The authors are not sure whether they understood this comment correctly. Sea-ice concentration retrieval algorithms have been developed to derive the sea-ice concentrations. Usually this is done such that one can retrieve the sea-ice concentration globally as good as possible at every time of the year. The question is not whether algorithms are designed to retrieve the ice-surface fraction or the sea-ice concentration but the question is how accurate the sea-ice concentration retrieved during summer still is and whether we really know what exactly is retrieved. If, during summer, a sea ice concentration algorithm provides 100% sea ice then it is not sure whether this is 100% with 30% melt ponds (and one gets 100% because the sea-ice concentration over the 70% ice-surface fraction is overestimated, or whether this is 100% for a REAL 70% sea-ice concentration with 30% REAL open water between the ice floes. This is not satisfying.

The authors would like to make also the point that users do not know what the different algorithms are retrieving during summer and that there is no way (currently) to state whether the summer-time SIC estimates are biased or not. What we wish to show is how to improve the situation that the SIC obtained complies with physics.

Still we agree with the reviewer that it is nice to show this inter-comparison and we now do and also actually use this as an argument why it is necessary to carry out further investigations. The Section 3.2, Figure 4, and Tables 5-7 refer to this issue now.

**The reviewer believes the author's work would have more impact in the future if more time is spent on refining and clarifying its presentation. As an overall impression, the reviewer found the results section quite dense and noted considerable redundancy in some discussion throughout the paper. The reviewer suggests authors try to con- solidate statements to their appropriate sections to reduce these redundancies. The reviewer also noted excessive detail in describing some sections of the results that was not paired with relevant analysis – this made some sections a bit rote. Some of these descriptions of the data could be reduced and organized following clear statements about what they show. The reviewer also strongly encourages having an editor go over the text. There are many punctuation errors (particularly dozens of missing commas)**

**and many instances of plural subjects with singular verb conjugations (i.e. the sentence on page 10, line 22-23) and several other odd wordings which may be hard for a non-native English speaker to eliminate.**

We thank the reviewer for these comments. We worked on the over-all structure of the manuscript, tried to avoid duplications and redundant sections, and tried to be more concise in descriptions and statements. Moreover, we worked on placing statements where they belong to. Actually, the manuscript has been completely re-written.

**There are lots of acryonyms being made up in this work (TB, ISF, SIC etc). The casual reader will not read the paper from end to end and/or may have different ideas of what these mean from prior works. As written, a thorough reading is required to find and becoming conversant in all these new acronyms. The reviewer strongly suggests a table of acronyms be created and placed into the document near the beginning. The reviewer also regularly became confused about the origin of particular data products. Since the key to the entire paper is a comparison of MODIS-derived vs. microwave- derived products, all MODIS derived products should be somehow clearly differenti- ated from all AMSR-E or SSM/I derived products in the acronyms (ISF, for example is MODIS-derived, but not denoted as such in a manner similar to MSIC). Perhaps all MODIS derived product acronyms would start with 'm'.**

We thank the reviewer and agree. In the revised manuscript we use acronyms only where we think they are needed. For instance we write "sea-ice concentration" and "brightness temperature" throughout the paper instead of using SIC and TB. We try to limit acronyms to the satellite sensors, to FYI and MYI, and to the algorithms applied to the microwave brightness temperatures. We are confident that this has increased the readability of the manuscript substantially.

**Specific comments**

**1. Several times it is mentioned that brightness changes in the sea ice surface itself may counteract some of the melt pond covering. Please quantify the relative magnitude of brightness changes compared to melt pond flooding.**

The plots showing AMSR SIC vs MODIS ISF for a number of algorithms show that for ISF=100% the AMSR SIC is often between 120 and 140%. This corresponds to an overestimation of SIC by 20-40%. As we describe in the revised version this overestimation is partly due to the melt ponds and partly due to other effects. We have re-written the introduction accordingly and make clear now that and how brightness temperatures change as a function of changing snow properties – see also new Tables 1-3.

**2. Page 2 line 18. These references are not the most appropriate for describing the physical processes of melt pond formation. Eicken et al., 2004; Polashenski et al., 2012; and Landy et al., 2014 are more focused on physical processes of pond formation. Perovich and Polashenski, 2012 is primarily focused on the evolution of albedo, as is Perovich et al., 2003. Petrich does discuss the connection between snow and pond locations.**

We thank the reviewer for these corrective actions. We re-wrote these sentences and changed the references accordingly.

**3. Page 2 line 22 – "can cover up to 50-90%" this is an un-necessarily sensationalist statement, particularly for a paper which is trying to quantify the TYPICAL impact of ponds on SIC retrieval. It would be more appropriate to discuss the TYPICAL coverage of melt ponds rather than the EXTREME bound. The references here are also not particularly relevant. Eicken is not primarily focused**

**on pond coverage but rather on the processes controlling ponding, papers published by Perovich in 2011 only reference other direct works on melt ponds. Yackel and Barber is appropriate, but only one of many. Also, Landy et al., 2014; Polashenski et al., 2012; Hanesiak and Barber etc. Futher, some of the references here and elsewhere are not found in the reference list (e.g. Perovich et al., 2011)**

We thank the reviewer for these clarifications. We re-wrote the sentence with the melt-pond fraction and change / extended citations / references accordingly.

**4. Page 2 line 25 – Albedo values certainly vary, but these albedo values are simply incorrect. Dry snow covered ice has an albedo of about 0.8. Bare, unponded melting FYI has a value of about 0.55 +- 0.1, depending on reference, and MY has 0.6+- 0.1. Melt pond covered ice tends to be lower than 0.5. Ponding therefore does not reduce ice albedo from 0.8 to 0.5, but rather from somewhere in the range 0.45-0.7 to somewhere in the range 0.1 to 0.5. Perovich, 2003 is a good reference for MYI, but does not reflect current state of the literature which increasingly works on FYI. Perovich and Polashenski, 2012 describes FYI ponds, as does Frey et al., 2014.**

Thank you. We re-wrote the sentence dealing with the albedo and changed the citations / references accordingly. We could not find the reference Frey et al., 2014, though.

**5. Page 3, line 7 – This discussion of noise is important as the justification for trying to tease out why SIC from passive microwave products might be 'right for the wrong reasons'. It is worded poorly, and hard to follow. The project name and reason for conducting it is also not so important and could be dropped. The key is that the paper is largely about understanding whether the passive microwave products are in- terpreting changes in pond coverage as a type of noise in the SIC record.**

We completely re-wrote the introduction and actually completely skipped that issue but gave two more easy to understand motivations. We didn't feel like "noise" is the right term here because what the algorithms exploit to retrieve SIC is the contrast between sea ice and open water in terms of the emissivity. And open water is not really "noise".

**6. Page 3 line 15 "Melt ponds are pools. . . " Redundant. this was already established and could be deleted.**

Agreed and changed accordingly. See also our reply to comment 5.

**7. Page 3 Line 16 - on penetration depth of passive microwaves. Passive microwaves are emitted from the sea ice and snow – as is correctly stated here. They are attenuated by liquid water. Discussing them as if the 'penetration depth' is limited would be language more appropriate to an active sensor with energetic waves PENETRATING from above. In this case waves EMITTED from below are being attenuated along the path to the sensor (because it goes through water). A novice reader could better understand that the ponds are attenuating a signal from below.**

We think that "penetration depth" is actually the correct term used to identify the layer from which most of the thermal radiation originates. We added this information to the revised manuscript.

**8. Page 5 MODIS data – Pond algorithm is executed using C5 data – several recent papers have suggested that significant uncorrected sensor degradation was present on C5 data (e.g Lyapustin et al., 2014). Though the degradation is only a few percent, it is not the same on all bands, meaning spectrally based algorithms can be significantly impacted. It would be useful to comment on whether this impacts the MODIS melt pond retrieval meaningfully.**

We use the MODIS sea-ice parameter data as it is (and as it was built before the Lyapustin paper). In sub-section 2.1.2 we discuss the melt-pond product, and that results might change if we would use a different MODIS version for the product. But the MODIS algorithm is not the subject of the current study. We cannot comment yet on whether issues with the MODIS data version are important for MPF dataset. We note, that at least for the present paper, a degradation of the quality of the MODIS data is less important because we focus on a three month period in one year only.

**9. MODIS data- It appears the locations you are focusing on, with high ice concentration, are all very far north. Here the MODIS data quality is likely to be quite poor because MODIS sun synchronous orbit does not place the sensor over high northern latitudes, and all retrievals are made at high off nadir angles, with low so- lar zenith angles, through long atmospheric path lengths. Under these conditions the MODIS surface reflectance products are well known to have substantial issues. The authors must at address this. The reviewer feels this is an important enough issue that the authors must examine whether the conclusions about SIC over representation by passive microwave method apply at lower latitudes where MODIS data is likely better. Such a comparison may reveal that the FYI /MYI differences are not actually cause by ice type. This may require restricting the timeframe evaluated if 100% ice coverage is required.**

We are aware of the limitation and refer here to our reply to the respective general comment of the reviewer. Also in the revised manuscript we do not focus on ice-type discriminations that much anymore.

**10. Page 6 Paragraph 3, bias correction – Where does the 3% global addition to the MSIC come from?**

We thought that by including Figure 2 and the respective discussion we have explained this issue enough. In order to better underline what we did and why, we have added more and also revised information in the respective section.

**11. Page 8 paragraph 3 – Ice age. It should be discussed and understood that the ice age within a 4 year cell is actually mostly less than 4 years, because all leads forming over the 4 year duration that at least some ice remained in the area refroze as younger ice. Many 4+year packs are composed of a large fraction of younger ice. As a result, this may not be a very effective mechanism for avoiding the influence of FYI. If the authors wish to really focus on the MYI/FYI differences they may consider using back trajectories to examine where the MYI was at the end of summer in the previous year, and eliminating MYI originating from areas of low ice concentration (where likely much FYI formed between the MYI floes).**

We actually thought that we have discussed this issue in depth in the previous version of the manuscript. However, since in the meantime a new version (V003) of the sea-ice age data set became available we repeated our analysis with the new data set. We seized the opportunity to also slightly change the way of our retrieval. Instead of computing an average age we now look into how the ice age varies in 100 km MODIS sea-ice parameter grid cell. We keep the ice age as is and count the number of ice-age data pixels (12.5 km x 12.5km) for each 7 x 7 pixel array co-located with the MODIS data. We subsequently assign the ice type "first-year ice" to the MODIS grid cell if > 90% of the pixels exhibits an ice age of 1. We assign the ice type "multiyear ice" to the MODIS grid cell if 90% of the pixels have an ice age of 3 or more. We note that by doing so, the number of MODIS grid cells assigned first-year ice increased while the number of MODIS grid cells assigned multiyear ice decreased. We note further – and also demonstrate this in Figure 1 – that the frequency with which first-year ice with a sea-ice concentration > 90% is sampled in our data set is quite small. There is a separate sub-section (2.4) about this data now.

**12. Page 9 line 14. Morassutti and LeDrew primarily show that depth of the melt pond is not causally related to spectral response in visible wavelengths, but rather related to the underlying ice properties. . . so the reference should be after the first clause, and the second should be deleted. Deeper ponds on MYI actually appear spectrally similar to shallow, early season ponds on FYI- again because the predominant factor is underlying ice properties.**

We thank the reviewer for this comment. Actually we have deleted that part from the MODIS data section because it did not add information about the quality of the currently used product.

**13. Uncertainties of MODIS sets – this section would benefit from a summary/concluding statement. In total, adding up all these errors in the MODIS sets, do they or do they not have the potential to alter the fundamental conclusions of this paper.**

We did add our conclusions about the uncertainties after revising the information about potential biases and inter-comparison studies (see sub-section 2.1.2).

**14. Line 23-25. This sentence would be better stated "Passive microwave emissions from the sea ice are attenuated within a path length of several mm of water, at the frequencies. . . hence in theory, melt ponds fully attenuate microwave emissions. . . and appear the same as leads.' Discussion of penetration again may confuse the novice reader into thinking this is some type of active sensing. Same comment applies to top of page 15 in section 4.1**

We still think penetration depth is the correct term – but see our reply to comment no. 7.

**15. Page 11 line 11 – these other factors impacting brightness temperature are very important and they are brought up repeatedly but never really addressed. This redundancy should be eliminated and a more complete discussion of them should be included. What is the range of expected impacts from surface wetting on TB? For example- can the authors show that these are secondary in magnitude to the pond impacts? The reviewer is left concerned that the increasing brightness temperature of wet and metamorphosing snow could offset pond reductions in TB – leaving SIC algorithms right for the wrong reasons.**

We have completely re-written the introduction and the discussion and added specific information about brightness temperature changes in response to snow property changes. We also devoted much of the discussion to snow property changes. We now actually use the different sensitivity to such property changes as the main argument why some algorithms are more sensitive to melt ponds than others.

**16. Page 12 last paragraph and page 13 first paragraph. Here would be a good place to quantify the TB change associated with these other changes.**

See our previous reply to comment 15.

**17. Section 4. This section is dense and challenging to get through for all but the most intrepid reader. Reviewer suggests it will have more impact if presented more concisely, perhaps with several tables displaying results as a matrix of algorithm with over/under prediction.**

We completely re-wrote the results and discussion sections.

**18. Why not plot MSIC against SIC – perhaps the algorithm is actually working for the wrong reasons.**

We added a section where we compare AMSR-E SIC against MODIS SIC (section 3.2, Figure 4, Tables 5-7).

**19. Page 17 Line 1- what gives the authors confidence that the range of bias in ISF does not exceed 10%, particularly given the small subset of the MODIS Melt pond data and extreme northern latitudes investigated? The reviewer is not convinced of this level of accuracy.**

We have re-written the MODIS sea-ice parameter quality discussion and devoted an own sub-section to that: 2.1.2.

**20. Page 19 Paragraph 2. This discussion about other factors strongly argues that this analysis should include investigation of smaller areas separately, so that impacts of melt timing can be considered. Last sentence of this paragraph is very long. The reviewer feels that such an analysis could greatly strengthen this work.**

We agree with the reviewer that it is often desirable to look into smaller areas to better understand differences between geophysical variables obtained with different approaches from satellite data. However, neither the main aim of the paper nor the number of co-located data warrants taking a look into smaller areas. For such studies our data set is simply too small.

We'd like to add that this daily MODIS melt-pond fraction data set was specifically derived for the ESA SICCI project; it is an opportunity data set. Investigations focusing on smaller areas could be done with the 8-day MODIS sea-ice parameter data set but this would be beyond the scope of this paper and will be topic of future work.

We note that by now showing plots of the temporal evolution of the melt signal we have partly satisfied the request of the reviewer.

**21. Section 4.3 Paragraph 1 and 2. This would seem to indicate that the NASA Team algorithm is behaving COR- RECTLY at its stated purpose – observing SIC. Perhaps for the wrong reasons, but nevertheless, the NASA team algorithm does not target ISF as the authors reason it should, it targets SIC. It is not accurate, therefore to state that ISF is overestimated by NASA_Team algorithm, because this is not what the algorithm purports to produce. A note of this must be made here. Also, how does the team algorithm work? Does it effectively include an empirical correction that is handling a presumed melt pond fraction?**

The NASA Team algorithm was developed (Cavalieri et al. 1984) based on SMMR observations in February 3-7, 1979. They use the fact that polarisation (18V and 37V channels) is substantially different for OW and ice. The tie-points were also retrieved from these observations, and later from SSM/I measurements (so that the tie-points are always updated with new instrument). But they are not varying with season. Therefore we do not think it can effectively include an empirical correction that is handling a presumed melt pond fraction, given that the microwave signatures of summer ice are different from those of winter (as shown in the manuscript). The latest NASA Team algorithm description is provided at NSIDC's web-pages (http://nsidc.org/data/docs/daac/nasateam/index.html), where it is clearly stated that melt ponds are indistinguishable from open water and the extent to which they affect summer sea ice concentrations is uncertain. With the revised version of the paper we shed some light into this last notion.

**22. Section 4.4. This section is un-enlightening. The authors describe the plots but fail to discuss what these results mean or why they occur.**

We did re-write the results and discussion sections completely. This section does not appear in this form in the revised manuscript.

**23. Page 22 line 28 – yes, but what would this reduction do to actual SIC – the parameter that the algorithm is designed to retrieve.**

In the revised version of the manuscript scatterplots of SIC retrieved from passive microwave algorithms versus SIC from MODIS will be shown. Here an example for the NASA Team algorithm is demonstrated.

[Figure]

The slope is close to 1. However, given that melt ponds are misinterpreted as open water by passive microwave algorithms (please see our arguments in the manuscript and the reply to the nr. 21), it is more relevant to look at the slopes in the scatterplots of SIC from passive microwave algorithms versus ice surface fraction (ISF) from MODIS (Figure 7 of the manuscript). In these plots the slope is larger than 1, which confirms the ISF overestimation by the algorithms. A situation when an algorithm gives the right SIC (for the wrong reasons) will only be the case for one particular MPF where the overestimation of ISF exactly balances the underestimation of SIC where melt ponds occur.

We think we have addressed these issues now in-depth by presenting comparisons between AMSR-E and MODIS sea-ice concentrations (section 3.2, Figure 4, Tables 5-7) and by the new discussion of the results which involves a discussion of the impact of snow and surface property variations.

**24. Page 23 line 21 – this reviewer is not convinced that the MODIS ISF and MPF have this low of a bias under the extreme circumstances of high latitude, high off nadir angle, low solar zenith angle in the study region.**

We refer the reviewer to sub-section 2.1.2 of the revised manuscript. In addition, comparison studies with aerial photography and also Comparison with the MERIS based melt pond data set from University of Bremen (unpublished Master thesis from H. Marks) show good correlations and therefore uncertainties of 10% are reliable. Additionally, as stated above, the MODIS data is atmospherically corrected and only data with the best quality score are used.

**25. Page 23 line 24 – The reviewer finds this statement to be theoretically accurate but poorly informed. While the addition of more channels could theoretically increase the number of surface types discriminated, the spectral signature of FY ponds + MY ponds as well as Bare and Snow covered sea ice overlap considerably. Additional channels are extremely unlikely to add orthogonal information in this system and further differentiation is unlikely.**

We absolutely agree with the reviewer that we should have elaborated more on potential possibilities to improve the MODIS sea-ice parameter data set. However, we felt that this issue is not topic of the paper and we left it out in the revised version of the manuscript. We note again that we used the data set developed with the method of Rösel et al. (2012) as it is without further optimizations. The development of an improved melt-pond fraction retrieval algorithm is not topic of this paper.

**26. Page 24 line 3 – this reviewer is not convinced that the ice type relationship comes entirely from the microwave side of the data. MODIS ISF is also likely to be impacted by ice type, both due to changes in the spectral character of MY ponds vs FY ponds and due to the MY ponds commonly being more deeply recessed into the ice surface, and therefore less visible at high of nadir angles of MODIS. Also, see next comment.**

There are two arguments against this. 1) we focus on July (and should maybe include data from June as well) where the spectral character of FYI and MYI melt ponds might not be that different yet because FYI has not yet melted through (or close to through); 2) we focus in July and there for melting into the sea ice isn't as advanced as later in the season (e.g. August).

We also doubt that MYI ponds are less visible – they might be deeper than FYI ponds, but the spectral signal is mainly driven by the thickness of the underlying ice, therefore, the FY and MY ponds COULD have kind of similar signature. Furthermore, the spectral curve used for the ANN training was an average curve for both types of ponds. With that we assumed to cover both types

However, we put much less weight on the ice-type issues in the revised manuscript and only show statistical values of the comparisons for completeness.

**27. Page 25 Line 29 – this is a very large majority of the data concentrated in MYI. This weakens the conclusions based on ice type considerably. Further, all the FYI is at a lower latitude, where the geometry of the satellite sensors is quite different. The reviewer feels that view geometry must be eliminated as a cause for the FYI MYI discrepancies if the authors are to retain discussion of FYI and MYI**

We put much less weight on the ice-type issues in the revised manuscript and only show statistical values of the comparisons for completeness.

**28. Page 26 Line 11-15 – this is approximately the 5th time these other factors are mentioned in the paper. A considerable redundancy. Further, the reviewer finds none of the discussions of these other factors sufficiently quantitative for the reader to assess whether they are so large in magnitude as to alter the paper's fundamental conclusions.**

We completely re-wrote introduction, results and discussion sections to make a better statement about this issue. We knew already that this could be the weakest point of our manuscript. This has, however, been corrected now.

**29. Page 27 Line 14 – this statement is the thesis of the paper. The reviewer does not believe it has been adequately supported yet, because the authors seem to be ignoring that the SIC retrieval algorithms are calibrated for SIC retrieval, NOT ISF – even though, theoretically, ISF is the response they see. Melt ponds SHOULD be interpreted as open water based on theory. The data however, actually does not indicate that the algorithms ARE interpreting the ponds as open water. The over estimation of ISF means that the value produced is actually closer to SIC. This would mean that the algorithms are NOT interpreting the ponds as open water. Further, the statement below noting that the current SIC algorithms produce SIC in winter and ISF in summer is also theoretically true but in practice unsupported by the data presented. The values do not represent ISF well. They are too high. They appear likely to represent SIC better. (Though such a comparison needs to be made) The reviewer believes the authors actually understand this distinction, but does not feel this has been clearly communicated to the reader yet.**

We refer the reviewer to the new discussion section of the revised manuscript.

**Page 26 lines 27-33 – Further comparison is needed here between MSIC and SIC. Perhaps the algorithm has empirically corrected for MPF impact, for all the wrong reasons. An empirical algorithm which did this would likely produce above-100% values for 100% ice cover, non-ponded. (ISF =100) Since these would be truncated to 100% by the user, this would not result in a SIC error in practice. Over**

**estimation could also be correct if this were a case of 100% ice cover with 30% ponds, because the algorithms are supposed to be retrieving SIC. In this case it appears likely that MODIS would retrieve an ISF =70% while several of the algorithms would find SIC = 100%. Again both could be correct at their design function, even if theory says the microwave derived SIC should be seeing something else.**

One of the main goals of this paper is to understand what passive microwave algorithms retrieve during summer melt, looking from a physically justified perspective. As it will be shown in the new figure demonstrating PMW SIC versus MODIS SIC (please see section 3.2 and Figure 4), the algorithms indeed produce correct SIC values, and therefore these products have been used for decades. However, such a performance is relying on an empirical adjustment, which will eventually cause inconsistency in the provided results. More specifically, our point is that if this might work for 30% MPF it will not give a reasonable result for 40% MPF (here we would get 90% SIC from such an algorithm)

Another example of wrongdoing is 80% real SIC=ISF. In this case of 30% overestimation the algorithm would give 110% (or perhaps 1.3*80%), which will be truncated to 100%. This is clearly wrong.

We believe that the way of thinking rooted in physics will provide not only better understanding of the measured parameters, but also can help to find a way of improving the usage of the retrievals. The only consistent way to do this is through ISF, since the passive microwave algorithms cannot distinguish melt-ponds from open water and we do not want to overestimate real ice concentration.

We refer the reviewer to the new discussion section of the revised manuscript.

References
Cavalieri, D. J., Gloersen, P., and Campbell, W. J.: Determination of sea ice parameters with the NIMBUS 7 SMMR, J. Geophys. Res., 89, D4, 5355–5369, 1984.
Rösel et al., 2012, A., Kaleschke, L., and Birnbaum, G.: Melt ponds on Arctic sea ice determined from MODIS satellite data using an artificial neural network, The Cryosphere, 6, 431-446, 2012.

**Referee #2 (Greg Flato)**

**Referee #2: G. Flato**

greg.flato@ec.gc.ca

**The ability to remotely sense and discriminate between sea-ice, open water and melt ponds is an important topic for model evaluation, process studies and initialization of operation forecast systems. This paper provides a very in-depth development and application of a scheme to estimate aspects of the area fraction of the surface types listed above.**

Thank you very much for your careful reading of our paper. We tried to keep the in-depth information in a clearer form in the revised manuscript which we re-wrote completely.

**Unfortunately, the paper is very difficult to read. It is so densely packed with abbreviations and acronyms and excess technical detail that it will be unintelligible to most readers. Even if one has a background in this field, the presentation style is a real impediment to effective communication. I would very strongly urge a complete re-write of the manuscript with an eye to simplification, clarification and more organized flow of material.**
We took this comment serious and completely re-wrote the manuscript.

We remove almost all of the acronyms to improve readability.

We structured the text much more than before. We start with a re-written introduction, have a data and methods section, where we go through the data sets one by one and have one extra sub-section in the section about MODIS data into which we put all the uncertainty discussion of the MODIS sea-ice parameters.

We completely removed plots and discussions about brightness temperatures and derived parameters (former figures 4 to 6). Instead we follow the comments of reviewer #1 and start with a comparison of AMSR-E and MODIS sea ice concentrations before we motivate why we use MODIS ice-surface fraction for which purpose. In the presentation and discussion of the results we try to keep the information easy and try to focus on the most important issues.

We focused on comparisons between AMSR-E sea-ice concentration and MODIS ice-surface fraction and discuss the results by means of discussing how some of the algorithms work. We add snow property changes to this discussion and give much less weight to the ice type discrimination which we show for completeness only – in form of tables.

We add the time dimension by additionally looking at the temporal evolution of brightness temperatures and contemporary MODIS ice-surface fractions to better understand and interpret our results.

We much more structured both the results as well as the discussion sections.

**The same issues arise with the figures which are, like the text, almost impenetrable.**
The authors believe that by removing old complex Figures 4-6 the revised manuscript has become easier to follow with less redundancies.

**I am convinced that the length of the text and the number of figures could be reduced by half, which would also allow individual figure panels to be increased to a readable size.**

We thank the reviewer for this comment. We find it however extremely difficult to cut down the length of the paper substantially without hiding too much important information.

We are showing an inter-comparison of SEVERAL algorithms. This requires showing a number of figures and also giving specific comments to these as they mostly reveal a NUMBER of differences.

In the new Figures 4 and 5, we reduced the number of images per figure by 2, i.e. show results from 6 instead of 8 algorithms. We also don't show figures anymore which are specifically valid for multiyear ice.

**This requires careful consideration of what the key messages/ conclusions are, and what is the essential material that must be presented in order to substantiate these. There is no point publishing a paper that no one can or will want to read.**

Thank you very much for this thoughtful comment. We agree. Still we find that with the new, completely rewritten manuscript with better structure and flow, clearer motivations of the steps carried out, and a clearer focus on the main issue, we can and should show the number of figures we show in the revised version. Actually, the new manuscript has 8 instead of 12 figures … but we submit 4 figures as supplementary material.

**I read and re-read the Conclusions section multiple times and I must say I am still not clear on what the real take-home message is. Certainly there is a lot of detail about uncertainties and their source and the differences between different algorithms. But the last paragraph basically just says melt pond fraction is confounded with open water fraction in summer (something that has been well known since the early days of seaice remote sensing), that users should be aware of this, and that there nothing at the moment that can be done about it. Given all the preceding detail, it is surprising that nothing is said regarding which algorithms are more or less reliable and how a user might make choices when faced with a particular problem or application, or indeed how an 'essential climate variable' might be constructed. The second-last paragraph of the paper seems to provide some commentary on different algorithms, but having read it several times, I still cannot glean any concrete guidance.**

We agree with the reviewer.

First of all we hope that the reviewer agrees with us that the material published in the paper warrants to write the conclusion in the form of a "Summary and Conclusions" section.

Secondly, we didn't and still don't see ourselves in the position to tell a user which of the algorithms investigated should or of should not be used and which is best.

One reason is that we look at only ONE summer. It is a case study.

The second one is, that we did not (and still don't) have a consistent set of summer sea-ice tie points to carry out this study and inter-comparison in a FAIR way. At least – and we are happy about this – we were able to use a consistent set of winter sea-ice tie points and thereby were able to mitigate differences between the algorithms by using published tie points which might have been retrieved under special conditions in different sea-ice regions.

As we show and discuss for the two Bootstrap algorithm modes, using the summer sea-ice tie points from the literature does not yield to an overwhelming improvement.

We give some clearer messages now about
a) which algorithm is particularly / not particularly sensitive to melt ponds,

b) how large are over- and under-estimation of the sea ice concentrations for the case where the open-water fraction is from open water between the ice floes and for the case where the open-water fraction is mainly from open water in form of melt ponds on the sea ice,

c) why the algorithms behave differently and

d) which algorithms are potentially better suited for optimization during summer than others.

**So, my conclusion is that this paper requires quite a bit of work, and I would recommend major revisions.**

The manuscript has been rewritten completely and we thank the reviewer again for pointing us to a number of key issues which we hope to have addressed properly in the revised version of the manuscript.

**V. Tikhonov**

**vtikhonov@asp.iki.rssi.ru**

**It was of interest to read the manuscript "The impact of melt ponds on summertime microwave brightness temperatures and sea ice concentrations" by S.Kern et al. The authors have conducted a very diligent and serious work. However, they state that from satellite microwave radiometer data it is impossible to distinguish open water between ice floes from open water in melt ponds on ice floes. I would refer the authors to a paper by Tikhonov et al. (2015). It presents a new algorithm called Variation Arctic/Antarctic Sea Ice Algorithm 2 (VASIA2) for determining sea ice concentration from satellite microwave radiometer data. VASIA2 not only retrieves ice concentration but also calculates ice areas covered by snow-water mixtures (SWM), including melted snow and melt ponds.**

**Tikhonov V.V., I. A. Repina, M. D. Raev, E. A. Sharkov, V. V. Ivanov, D. A. Boyarskii, T. A. Alexeeva, N. Yu. Komarova. A physical algorithm to measure sea ice concentration from passive microwave remote sensing data. // Advances in Space Research. 2015. V. 56. N 8. P. 1578-1589. DOI: 10.1016/j.asr.2015.07.009.**

**Please also note the supplement to this comment:**

**http://www.the-cryosphere-discuss.net/tc-2015-202/tc-2015-202-SC1-supplement.pdf**

**Interactive comment on The Cryosphere Discuss., doi:10.5194/tc-2015-202, 2016.**

Dear Vasily Tiknonov,

Thank you for your comment to our melt pond paper.

We have implemented the VASIA2 algorithm (including VASIA) according to the specifications given in the Tiknonov et al (2015) paper referred to, and we have tested the algorithm on the MODIS melt pond fraction data in our reference data package (Round Robin Data Package). Figures 1-4 show the results. All data have an ice concentration >90% according to the MODIS data processed according to Rösel et al, 2013.

A number of conclusions can be drawn:
1. In the MODIS data the evolution of melt ponding (melt pond fraction) increases slowly in June, peaks in July with values between 20 and 35%, and is quite stable in most of August around 15%
2. Many VASIA ice concentrations, especially in July, are between 40 and 60%.
3. Only very few VASIA SWM reach above 15% and those that do are typically larger than 40%
4. The relationship between VASIA SWM and MODIS MPF is not convincing, and most data with melt ponds according to MODIS result in VASIA SWM of 0%.
5. If SWM includes (as claimed) wet snow as well as melt ponds we would expect SWM values that are larger than the MODIS melt pond fractions in our reference dataset. This is not the case. On the contrary most SWM values are 0 or very close to 0.

Based on the above findings we do not see that the VASIA2 algorithm calculates ice areas covered by snow-water mixtures (SWM), including melted snow and ice as stated in your comment.

[Figure]

Figure 1. MODIS sea ice concentration (MSIC) and melt pond fraction (MPF). Only those measurements are selected where MSIC > 90% and cloud fraction < 5%. X-axis is just an observation counter. Vertical blue lines indicate July 1 and August 1.

[Figure]

Figure 2. VASIA2 sea ice concentration (SIC) and snow water mixture (SWM). The data are filtered the same way as in Fig. 1: only those measurements are selected where MODIS sea ice concentration (MSIC) > 90% and cloud fraction < 5%. X-axis is just an observation counter. Vertical blue lines indicate July 1 and August 1.

[Figure]

Figure 3. Scatterplot of snow water mixture (SWM) as obtained by the VASIA2 algorithm and melt pond fraction (MPF) obtained from MODIS. The red line indicates 1-to-1 line.

[Figure]

Figure 4. Scatterplot of sea ice concentration (SIC) obtained by the VASIA2 algorithm and by MODIS (MSIC).

---

## Author Response (AR2)

**Referee #3, Georg Heygster**

**Review of TCD manuscript Kern et al. The impact of melt ponds…**

The manuscript constructs a data set of 100% sea ice concentrations (SIC) from the summer 2009 based on optical observations (MODIS) and analyzes the influence of varying melt pond fraction (MPF) on the satellite observed microwave brightness temperatures (TBs) and resulting SIC retrievals.

The manuscript treats an important subject in an exhaustive way, and also development in time of TB and SIC is considered. I suggest to accept the manuscript after small changes.

Title and abstract well reflect the content.

My main comments are:

The sea ice and melt pond concentrations based on the MODIS data are considered as ground truth. It should be mentioned that this assumption is the base of the complete investigation (in fact it is the best we currently have). However, the MODIS based retrieval is a static one which is not able to reflect potential developments of the optical melt pond signatures over the melting season (The MODIS retrieval algorithm does not have tie points but its retrieval is based on a neural network described by a larger set of parameters). This point is mentioned in the last paragraph of P8 (first two sentences), but it should be added that this is a central assumption of all subsequent analyses.

The paragraph continues 'The spectral properties of melt ponds are likely to approach those of leads and openings as melt season progresses.' This may be true for ponds on first-year ice, but it is not the case for blue ponds which typically appear on multi-year ice where the ice below the pond is thick. This aspect will not cancel the complete manuscript (the used data are the best we have

**currently), but this assumption should be mentioned. It is essential for the (except this point excellent) discussion of the temporal development in Figs. 7 and 8.**

**The subsequent part of the same paragraph states that a misclassification between melt ponds and open water would have no implications for the net sea-ice surface fraction. It would affect only the melt pond or the open-water fraction. This is correct for the net sea-ice surface fraction. However, an error in the open water fraction would also affect the MODIS sea ice concentration which is used in Fig. 4. Should be mentioned.**

We do really appreciate the work of the reviewer taking the time to read our manuscript carefully. We thank the reviewer in particular for pointing us to some critical information missing in the above three paragraphs. We have changed the text accordingly as follows:

"More important is the potential misclassification of one of the surface types. The reflectance values used are fixed for the entire summer season and the entire Arctic domain. Therefore the MODIS sea-ice parameter retrieval does not account for the spatiotemporal variability in the spectral properties of the melt ponds or the non-ponded sea ice. These spectral properties change as a function of ice type and melt-season duration. The spectral properties of melt ponds on first-year ice are likely to approach those of leads and openings as melt season progresses while for melt ponds on multiyear ice these change less due its larger thickness and different internal structure. This could result in an over-estimation of the melt-pond fraction relative to the open-water fraction for first-year ice or vice versa, because the spectral space between sea ice and water is larger than between melt ponds and open water (leads). Such a misclassification would have, however, no implications for the net sea-ice surface fraction. It would affect only the melt pond or the open-water fraction. Therefore such a misclassification is likely not influencing the main results of the present paper but should be kept in mind when interpreting MODIS sea-ice concentrations (see sub-section 3.2)."

and further:

At the time of our analysis and writing this MODIS product was the best we could have, despite the above-mentioned limitations due to cloud cover and spatiotemporal variation of the ice-type dependent spectral properties of the summer sea-ice cover. The results of our quality analysis and the results of Marks (2015) confirm that we can take the MODIS sea-ice parameters as kind of the ground truth against which we compare brightness temperatures and sea-ice concentrations in Sections 3 and 4.

**In Figure 6 (which I find impressively instructive and rich of information), among other the parameter space of the Bootstrap_f algorithm is shown, which is used in the Bootstrap algorithm for low ice concentrations only. It might be more relevant to show the parameter space diagram for an algorithm which claims to work at high ice concentrations. If you keep showing Bootstrap_f, then explain the reason in the manuscript.**
**The same comment applies to Fig. 8 (Bootstrap_f) which could be replaced by Fig. S01 (Bootstrap_p) and it applies to the whole Section 4.3.2.**

Although we agree that at first glance Bootstrap_f is not used at high ice concentrations we would like to point the reviewer to a few key points.

First of all, both modes of the Bootstrap algorithm are widely used in the community. It is hence a well-known algorithm which needs to be shown here. We feel that we do not repeat this information in the manuscript.

Secondly, we feel that it is important to show both modes of the algorithm because the threshold sea-ice concentration at which the sea-ice concentration retrieval switches between the two modes depends on season and ice type – even though it seems to be close to 90%. Also, we feel that it is an important piece of information to show whether for a particular sea-ice situation one of the modes provides a biased sea-ice concentration value while the other does not.

Third, the two other hybrid algorithms used, OSISAF and SICCI either use Bootstrap_f or CalVal (which is identical to Bootstrap_f) for low ice concentrations. While it might be less interesting to consider the Bootstrap_f mode in case of the OSISAF algorithm, for which the switch between Bootstrap_f and Bristol (used at high ice concentrations) is occurring in the range 40% to 80%, it is more interesting and relevant to consider the Bootstrap_f mode for the SICCI algorithm because the above-mentioned switch occurs in the range 70% to 90%. Actually we have added the thresholds where the switches occur in sub-section 2.3 on page 10: "For the high sea-ice concentrations we focus on in this paper, these two hybrid algorithms are almost identical to the algorithm, which is employed at high sea-ice concentrations, that is Bristol in case of the OSI-SAF (zero weight at < 40%, full weight at ≥ 80%) and SICCI (zero weight at < 70%, full weight at ≥ 90%) algorithms and Bootstrap_p in case of the Comiso Bootstrap algorithm."

Finally, AMSR-E sea-ice concentrations retrieved with the two modes of the Bootstrap algorithm show the lowest and highest sensitivity to the melt-pond fraction (see page 15, first paragraph of Section 4) which we think by itself is a good enough reason to focus as much on the Bootstrap_f mode as we focus on the Bootstrap_p mode. Again, since we give this information in the text already we don't think that we need to argument more for why we include the Bootstrap_f mode.

**Detail comments:**

**Section headings 3 (Results?) and 4 (Discussion?) missing**

We added the missing section headings like the reviewer did suggest.

**Have consistent denoting of supplement figures, not mixing e.g. S1 and S01.**

We found that supplement figures are denoted S1 to S4 in the text but S01 to S04 in the supplements document. We have changed the latter to comply with the main manuscript.

**Abstract P(age) 1 L(ine)16: consider replacing 'between the melt ponds' by 'starting well before formation of melt ponds when temperatures are still below zero' or so.**

We don't agree with this suggestion. The reviewer seems to have understood that we are referring to snow and sea-ice properties before melt did commence. In fact, we are referring to the sea ice between the melt ponds (locally). The point we wish to address in this paper is that not only the pure presence of melt ponds, i.e. open water on top of sea ice, but also and in particular the properties of the sea ice (with snow or bare) between the melt puddles change and cause (apparently) exactly those changes in surface emissivity which makes ponded sea ice to look like it is 100% sea ice without

liquid water on top. In order to avoid confusion we changed "between the melt ponds" to "itself" to make clear that we refer to the sea ice properties.

**P2L14 since over 35 years -> for more than 35 years**

Changes as suggested.

**P5 L15 delete 'JGR-C'**
We removed JGR-C and other journal abbreviations which we left accidentally in the text of the previous version of the manuscript.

**P7 L24 'We use only reflectance values with the highest quality score. This ensures that cloudy pixels and pixels with cloud shadows, pixels with sun zenith angles > 85° and pixels with sensor viewing angles > 60°, plus pixels complying with four additional flag criteria are not used.':**
**- scores not defined. No need to mention because criteria are described in the subsequent text. Omit 'score'?**
**- where from cloud shadows known? Visual inspection?**
**- 'four additional flags': which ones?**
We deleted "score". We are more specific with the quality flags now and added all to the manuscript so that these sentences now read: "We use only reflectance values with the highest quality. This ensures that cloudy pixels and pixels with cloud shadows, pixels with sun zenith angles > 85° and pixels with sensor viewing angles > 60°, data from faulty or poorly corrected L1B pixels, pixels containing the default or the highest aerosol level and pixels without any correction for the atmospheric influence are not used."
We refer the reviewer to the documentation of the MODIS data set for the details regarding how these flags are derived, e.g. cloud shadows.

**P8 L11 'we set the few negative melt-pond fractions ...to zero: Were other surface type concentrations also changed in order to keep them summing up to 1?**
Yes. This is evident from Figure 3 where the net open water fraction and the net ice surface fraction add to 100%.

**P9 L25 'Only data with footprints which center I located within 5 km...' Unclear. Do you mean 'Only data with footprints with centers located within 5 km...'?**
Yes. We have changed the text accordingly.

**P17 L28 '(5-day periods)': shift to first occurrence of pentad on P 12.**
We moved 5-day periods to page 12 now. The first sentence where "pentad" occurs reads now as follows:  "Then the melt-pond fraction starts to increase, first slowly: days 20-30 (5th and 6th 5-day period or pentad of June), then rapidly: days 30-45 (1st to 3rd pentad of July)."

**P18 L32 '... and the MODIS ice-surface fraction is over-estimated.': where from known? I cannot read this from Fig. 8a or 6c. I only read from Fig. 8b that Bootstrap_f will find SIC>100%, not MODIS. Explain.**
This was a misunderstanding borne out of sub-optimal formulation. The sentence now reads: "Most Bootstrap_f sea-ice concentrations exceed 100% and over-estimate the MODIS ice-surface fraction."

**P20 L1 'at this side of the parameter space': meaning unclear. Which side? Of what? Which figure?**

We have changed two sentences to make clearer what we are referring to. These two sentences now read: "The distance between the cyan line and the winter ice lines in proximity to the FYI tie point, measured along the dashed white line (Figure 6 c), suggests that we would reduce Bootstrap_f sea-ice concentrations by 10% to 15%. Therefore, at the FYI side of the parameter space, Bootstrap_f sea-ice concentrations would be ~100%."

**Reference Tschudi et al. (2016): Replace '[indicate subset used]' appropriately.**
We added the respective information.

In addition, we carried out a spell check of the manuscript and went through it, deleting one or two superfluous half-sentences, and tried to be clearer where we thought additional information might be needed. We also added one reference.

[revised manuscript text omitted]